# *DSCAM* gene triplication causes excessive GABAergic synapses in the neocortex in Down syndrome mouse models

Hao Liu[1,2�], René N. Caballero-Florán[3�], Ty Hergenreder[1�], Tao Yang[1], Jacob M. Hull[3], Geng Pan[1], Ruonan Li[1], Macy W. Veling[1], Lori L. Isom[3], Kenneth Y. Kwan[4,5], Z. Josh Huang[6,7], Peter G. Fuerst[8], Paul M. Jenkins[3,9]*, Bing Ye [1,2‡]*

**1** Life Sciences Institute, University of Michigan, Ann Arbor, Michigan, United States of America, **2** Department of Cell and Developmental Biology, University of Michigan, Ann Arbor, Michigan, United States of America, **3** Department of Pharmacology, University of Michigan Medical School, Ann Arbor, Michigan, United States of America, **4** Michigan Neuroscience Institute, University of Michigan Medical School, Ann Arbor, Michigan, United States of America, **5** Department of Human Genetics, University of Michigan Medical School, Ann Arbor, Michigan, United States of America, **6** Department of Neurobiology, Duke University Medical Center, Durham, North Carolina, United States of America, **7** Department of Biomedical Engineering, Duke University Pratt School of Engineering, Durham, North Carolina, United States of America, **8** University of Idaho, Department of Biological Sciences, Moscow, Idaho, United States of America, **9** Department of Psychiatry, University of Michigan Medical School, Ann Arbor, Michigan, United States of America

☦ These authors contributed equally to this work.
‡ Lead contact.
* pjenkins@umich.edu (PMJ); bingye@umich.edu (BY)

**Data Availability Statement:** The data that underlie the figures are publicly accessible in https://doi.org/10.5281/zenodo.7714234.

## Abstract

Down syndrome (DS) is caused by the trisomy of human chromosome 21 (HSA21). A major challenge in DS research is to identify the HSA21 genes that cause specific symptoms. Down syndrome cell adhesion molecule (DSCAM) is encoded by a HSA21 gene. Previous studies have shown that the protein level of the *Drosophila* homolog of DSCAM determines the size of presynaptic terminals. However, whether the triplication of *DSCAM* contributes to presynaptic development in DS remains unknown. Here, we show that DSCAM levels regulate GABAergic synapses formed on neocortical pyramidal neurons (PyNs). In the Ts65Dn mouse model for DS, where DSCAM is overexpressed due to *DSCAM* triplication, GABAergic innervation of PyNs by basket and chandelier interneurons is increased. Genetic normalization of DSCAM expression rescues the excessive GABAergic innervations and the increased inhibition of PyNs. Conversely, loss of *DSCAM* impairs GABAergic synapse development and function. These findings demonstrate excessive GABAergic innervation and synaptic transmission in the neocortex of DS mouse models and identify DSCAM overexpression as the cause. They also implicate dysregulated DSCAM levels as a potential pathogenic driver in related neurological disorders.

## Introduction

Down syndrome (DS) is caused by an extra copy of human chromosome 21 (HSA21), and people with DS exhibit a number of medical conditions. A major challenge in DS research is to

**Funding:** This work was supported by grants from the National Institutes of Health (R21NS094091, R01MH112669, and R01EB028159 to B.Y.; R37NS076752 to LLI), a seed grant from Brain Research Foundation to B.Y., a grant from the Protein Folding Disease Initiative of the University of Michigan to B.Y., startup funds from the University of Michigan Department of Pharmacology to P.M.J., and a University of Michigan Rackham Merit Fellowship and a MI-BRAIN Predoctoral Fellowship to J.M.H.. The funders had no role in study design, data collection and analysis, decision to publish, or preparation of the manuscript.

**Competing interests:** The authors have declared that no competing interests exist.

**Abbreviations:** ACC, anterior cingulate cortex; ACSF, artificial cerebrospinal fluid; AIS, axon initial segment; APP, amyloid precursor protein; ASD, autism spectrum disorder; ChC, chandelier cell; CNV, copy number variant; DS, Down syndrome; DSCAM, Down syndrome cell adhesion molecule; HSA21, human chromosome 21; mIPSC, miniature inhibitory postsynaptic current; PFA, paraformaldehyde; PyN, pyramidal neuron; P28, postnatal day 28; RT, room temperature; sIPSC, spontaneous inhibitory postsynaptic current; SNV, single-nucleotide variant.

identify the genes on HSA21 that cause specific cellular and system alterations leading to the symptoms. Down syndrome cell adhesion molecule (DSCAM) is an evolutionarily conserved type I transmembrane protein encoded by a HSA21 gene [1]. In humans, the *DSCAM* gene resides in the DS critical region of HSA21 [1], which is associated with many symptoms of DS. We previously showed that protein levels of the *Drosophila* homolog of DSCAM, Dscam [2], determine the sizes of presynaptic terminals in sensory neurons without requiring the ectodomain diversity of the *Drosophila Dscam* gene [3]. Moreover, others reported that overexpression of *Dscam* impairs synaptic targeting and transmission in *Drosophila* [4,5]. These findings suggest that dysregulated DSCAM levels might contribute to neuronal defects in brain disorders in humans. In fact, altered DSCAM levels have been reported in multiple brain disorders, including DS [6], autism spectrum disorders (ASDs) [7–9], intractable epilepsy [10], bipolar disorder [11], and possibly Fragile X syndrome [3,4,12,13]. Although recent findings suggest a conserved role of *DSCAM* in promoting presynaptic growth in vertebrates [14,15], whether dysregulated DSCAM expression results in neuronal defects in brain disorders remains to be empirically determined.

In this work, we sought to determine the effects of altered DSCAM levels in mouse models of DS, in which the *DSCAM* gene is triplicated. Previous studies have shown that enhanced GABAergic inhibition impairs cognition in Ts65Dn mice [16–20], the most widely used DS animal model [21,22]. Furthermore, Ts65Dn mice show excessive GABAergic inhibition in the hippocampus [16,23–31]. Overproduction of GABAergic neurons caused by triplication of *Olig1* and *Olig2* contributes to excessive inhibition in the hippocampus [29–31]. However, several lines of evidence suggest heterogeneous etiology in DS brain disorders such that different brain regions exhibit distinct molecular, cellular, and physiological defects. For example, the frequency of miniature inhibitory postsynaptic currents (mIPSCs) is increased in the dentate gyrus, but not the CA1 region, of the hippocampus in Ts65Dn mice [25,31]. In contrast to extensive reports on GABAergic deficits in the hippocampus of DS animal models, very little is known about whether GABAergic signaling is altered in the neocortex. Previous studies showed that the sizes of inhibitory synaptic boutons are enlarged in the neocortex of Ts65Dn mice, suggesting possible alterations in GABAergic synaptic functions in this region [32,33]. In the present study, we show excessive GABAergic innervation of and synaptic transmission to neocortical pyramidal neurons (PyNs) in Ts65Dn and demonstrate that DSCAM overexpression in GABAergic neurons plays a key role in this process.

By combining genetic tools for sparse labeling and whole-cell patch-clamp recording, we found excessive GABAergic innervation and inhibition of cortical PyNs by chandelier cells (ChC) and basket cells in Ts65Dn mice. Genetic normalization of DSCAM levels rescued presynaptic overgrowth and excessive synaptic transmission of GABAergic neurons. Consistently, loss of *DSCAM* impaired ChC presynaptic growth and reduced inhibition of PyNs. In addition, we found that ChC axon terminal growth and bouton number are coupled and positively correlated with DSCAM levels in both wild-type and Ts65Dn mice. These findings thus highlight the critical role of DSCAM levels in regulating GABAergic synapses in the neocortex. Therefore, dysregulated DSCAM expression levels may be a common contributor to GABAergic dysfunctions in associated neurological diseases.

## Results

### DSCAM overexpression in Ts65Dn mice increases the number of GABAergic boutons on PyN somas in the neocortex

DSCAM is overexpressed in the brains of DS patients [6]. We found that it is also overexpressed in the Ts65Dn mouse (Fig 1A), a widely used DS model in which *DSCAM* is present in

3 copies [21]. Previous studies in Ts65Dn neocortices have demonstrated increased numbers of GABAergic synapses [34] and enlargement of these synapses [32,33], but which HSA21 gene(s) causes changes in GABAergic synapses is poorly understood. DSCAM is expressed in GABAergic neurons [35]. Thus, we asked whether GABAergic synapse number is altered in the Ts65Dn mice and, if it is, whether DSCAM is responsible.

Crossing female Ts65Dn mice with the male loss-of-function mutant of a gene of interest (e.g., *DSCAM*) yields trisomic mice with 2 functional copies of that gene. If heterozygous mutant is used, the progeny also includes the regular trisomic and euploid littermates (Fig 1B). As such, the contribution of a specific gene that is triplicated in Ts65Dn mice can be determined by comparing the mice that are "normalized" for the gene of interest with the trisomic mice [31,36]. To test the effects of *DSCAM* triplication by using this genetic scheme, we used a protein-null allele $DSCAM^{2j}$ [37–39] (referred to as $DSCAM^{-/-}$ here on). In Ts65Dn mice normalized for *DSCAM* gene dosage (Ts65Dn:*DSCAM*+/+/− genotype), the average level of DSCAM proteins were statistically indistinguishable from the euploid mice (Fig 1A). As a control, normalizing *DSCAM* gene dosage did not change the increased level of amyloid precursor protein (APP) in Ts65Dn (S1A Fig), which is encoded by another HSA21 gene that is important for brain disorders of DS [36].

Basket cells are the most common cortical interneurons that preferentially innervate the soma and proximal dendrites of PyNs [40,41]. We analyzed the GABAergic boutons formed on PyN soma (i.e., perisomatic boutons)—which are predominantly formed by basket cells—in cortical layer II/III of the anterior cingulate cortex (ACC) on postnatal day 28 (P28), a time after basket cell development is largely complete [42]. To visualize perisomatic GABAergic boutons, brain sections were triply labeled with anti-Bassoon (for presynaptic active zone) [43], anti-VGAT (for presynaptic GABAergic boutons) [44], and anti-GRASP1 (for PyN soma and proximal dendrites) [45]. We determined whether the number of GABAergic boutons around each PyN soma—as indicated by Bassoon+ puncta that overlapped with VGAT+ signals (S1B Fig)—was changed by the loss of *DSCAM* gene. Quantification was performed in a double-blind fashion to avoid experimenter bias.

Consistent with previous reports [33,34], we found that the average number of GABAergic boutons around each PyN soma was significantly increased in Ts65Dn in both ACC (Fig 1C and 1E) and the somatosensory cortex (S2 Fig). Strikingly, normalizing DSCAM expression level completely rescued the increased number of boutons. Average soma size of PyNs was not affected in Ts65Dn or Ts65Dn:*DSCAM*+/+/− mice (S1D Fig). These data suggest that the DSCAM overexpression in Ts65Dn mice increases the number of GABAergic boutons on PyN somas.

DSCAM is expressed in GABAergic neurons [35]. Next, we determined whether the extra copy of *DSCAM* gene within GABAergic neurons leads to the excessive GABAergic boutons on PyN soma. Previous studies in *Drosophila* sensory neurons [3] and mouse retinal ganglion cells [15] suggest a cell-autonomous role of *DSCAM* in promoting axonal growth. To address this question, we took advantage of *Lhx6*-Cre mouse line, which targets GABAergic interneurons, including basket and ChC cells, at early developmental stages [46,47]. Crossing *Lhx6*-Cre mice with mice carrying a floxed *DSCAM* allele ($DSCAM^{flox}$) [48] in the Ts65Dn background leads to the normalization of *DSCAM* gene expression specifically in GABAergic neurons. Normalizing *DSCAM* dosage in GABAergic neurons reduced the number of perisomatic GABAergic boutons to the euploid level (Fig 1D and 1F). These results suggest that the extra copy of *DSCAM* in GABAergic neurons leads to the excessive GABAergic boutons on PyN somas.

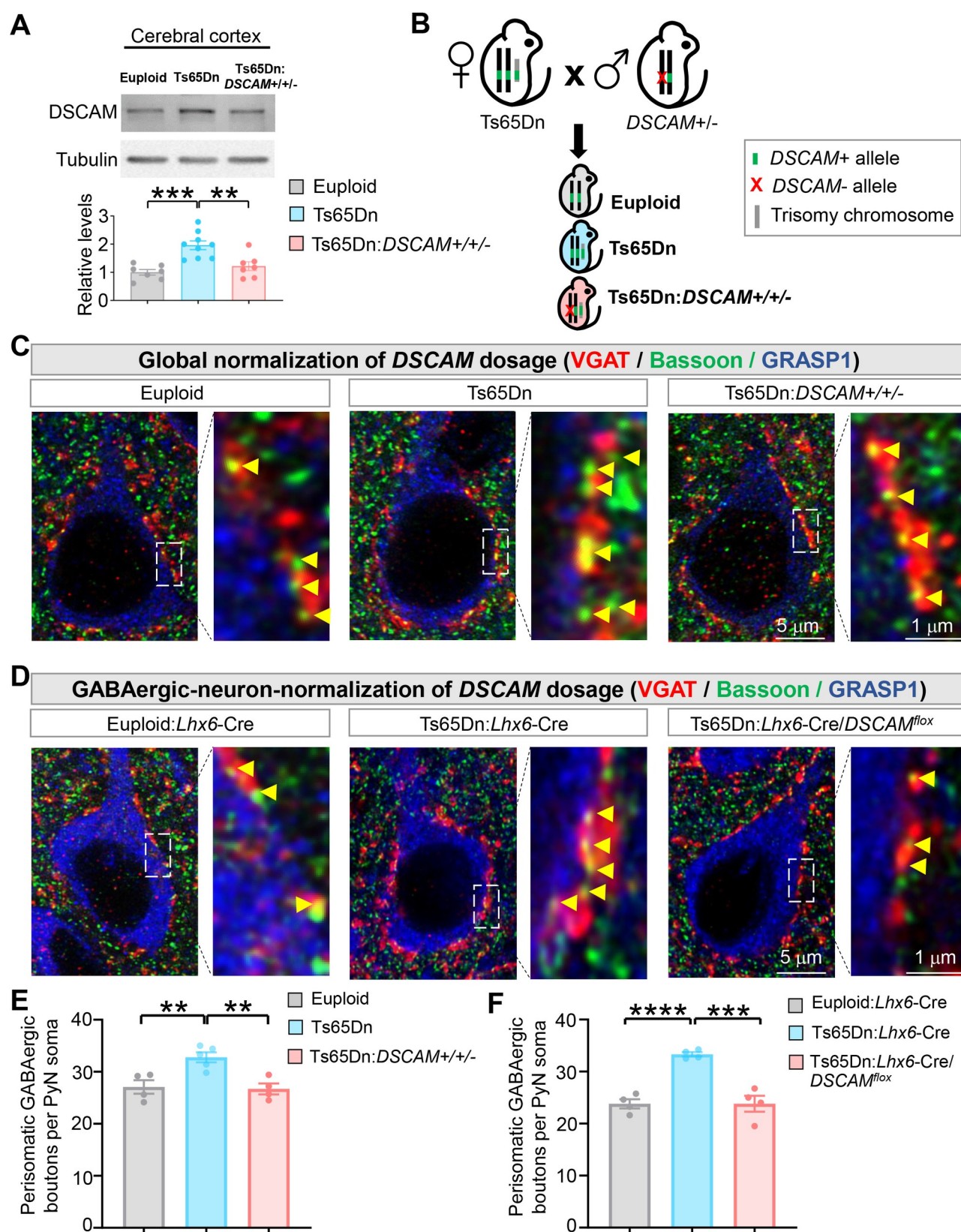

**Fig 1. Genetic normalization of DSCAM levels rescues the number of GABAergic boutons formed on PyN somas in Ts65Dn mice.** (**A**) DSCAM overexpression is normalized to the euploid level in Ts65Dn mice by introducing the *DSCAM²ʲ* loss-of-function allele. Euploid, Ts65Dn, and Ts65Dn: *DSCAM+/+/−* mice were obtained by crossing female Ts65Dn mice with the male *DSCAM²ʲ*. Shown are representative western blots (top) and quantifications (bottom) of neocortical samples from each indicated genotype. Each dot in the bar chart represents the sample from 1 mouse. (**B**) A schematic of the procedure that produced the mice for the experiments. (**C**) Representative images of perisomatic GABAergic boutons innervating PyNs in layer II/III of the ACC of euploid (wild-type), Ts65Dn, and Ts65Dn:*DSCAM+/+/−* (i.e., global normalization of *DSCAM* dosage). The right panel in each genotype group is the magnified view of the regions boxed by dotted lines in the left panel. The soma and proximal dendrites of PyNs were labeled by GRASP1. Yellow arrowheads point to GABAergic boutons as indicated by Bassoon+ puncta that overlapped with VGAT+ puncta. (**D**) Representative images of perisomatic GABAergic boutons innervating PyNs in layer II/III of the ACC of euploid:*Lhx6*-Cre+/−; (2) Ts65Dn:*Lhx6*-Cre +/−; and (3) Ts65Dn:*Lhx6*-Cre+/−/*DSCAMᶠˡᵒˣ* (i.e., GABAergic-neuron normalization of *DSCAM* dosage). (**E, F**) Quantification of the number of perisomatic GABAergic boutons per PyN in global (**E**) and GABAergic-neuron (**F**) normalization of *DSCAM* dosage experiments. For each mouse, 5–7 PyNs were analyzed; each data point in the chart represents the mean in 1 mouse. Unless specified, mean ± SEM is shown in the figures, and the statistical tests are one-way ANOVA for multi-group comparisons and post hoc Student *t* tests for pair-wise comparisons. **: $p < 0.01$; ***: $p < 0.001$; ****: $p < 0.0001$. The data underlying this Figure can be found in https://doi.org/10.5281/zenodo.7714234. ACC, anterior cingulate cortex; DSCAM, Down syndrome cell adhesion molecule; PyN, pyramidal neuron.

## Normalizing DSCAM levels rescues the excessive GABAergic synaptic transmission in the Ts65Dn neocortex

Despite previous reports of increased numbers and enlargement of GABAergic synapses in Ts65Dn neocortices [32–34], whether GABAergic synaptic transmission in the neocortex is increased in these mice remains to be determined. Our finding that the overexpressed DSCAM in Ts65Dn mice increased GABAergic boutons in the neocortex prompted us to test this possibility.

We examined GABAergic synaptic transmission in acute neocortical slices, using the whole-cell patch-clamp to record from PyNs in layers II/III of the ACC. We found that mIPSC frequency was increased by approximately 63% in Ts65Dn mice compared to euploid litter-mates (Fig 2A and 2B), which is consistent with increased bouton numbers in basket cells (Fig 1C and 1E). By contrast, we found no difference in mIPSC amplitudes between euploid litter-mates and Ts65Dn mice (Fig 2A and 2C), suggesting that postsynaptic responses may not be affected by the trisomy. Consistent with the role of DSCAM in the increased number of GABAergic boutons in Ts65Dn mice (Fig 1C and 1E), normalizing DSCAM expression pre-vented the increase in mIPSC frequency in these mice (Fig 2A and 2B), suggesting that DSCAM overexpression causes excessive GABAergic synaptic transmission. Similar changes in the frequency, but not the amplitude, were observed in spontaneous inhibitory postsynaptic currents (sIPSCs) among euploid, Ts65Dn, and Ts65Dn:*DSCAM+/+/−* mice (S3 Fig), consis-tent with increased GABA synaptic sites.

The resting potential, threshold, and action potential amplitude of PyNs were indistinguish-able among euploid, Ts65Dn, and Ts65Dn:*DSCAM+/+/−* mice (S4A–S4C Fig). We observed a subtle difference in the action potential half-width between Ts65Dn and euploid (S4D Fig); however, this slight change did not affect the depolarization and repolarization velocities (dv/ dt) (S4E and S4F Fig), which were not different among the 3 groups. These results suggest that the membrane properties of PyNs are not affected by trisomy or trisomy with 2 copies of *DSCAM*. Moreover, the firing frequencies and rheobase (defined as the minimal current required to evoke an action potential) were indistinguishable among euploid, Ts65Dn, and Ts65Dn:*DSCAM+/+/−* mice (S4G and S4H Fig). This suggests that the excitability of PyNs is largely unaffected by the trisomy or trisomy with 2 copies of *DSCAM*.

We further determined whether normalizing *DSCAM* dosage specifically in GABAergic neurons rescues the increased frequencies of mIPSC seen in the Ts65Dn neocortex. *Lhx6*-Cre mice were crossed with *DSCAMᶠˡᵒˣ* mice to generate *Lhx6*-Cre:*DSCAMᶠˡᵒˣ* mice in the Ts65Dn background (Ts65Dn:*Lhx6*-Cre:*DSCAMᶠˡᵒˣ*). Normalizing *DSCAM* dosage in GABAergic neu-rons in Ts65Dn mice completely rescued mIPSC frequencies to the euploid level (Fig 2D and

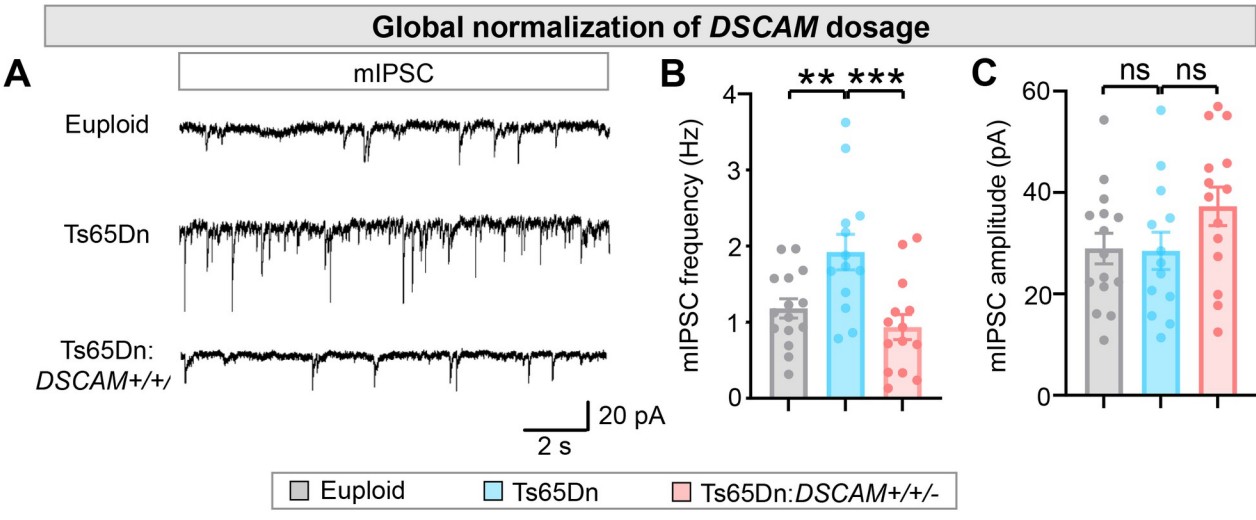

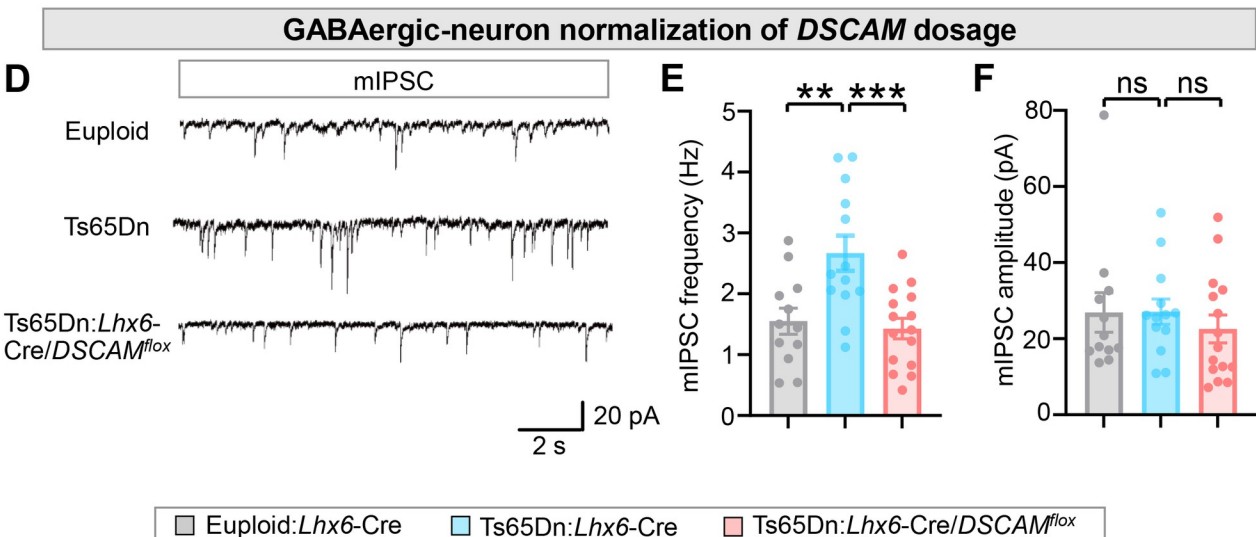

**Fig 2. Normalizing DSCAM levels rescues the enhanced GABAergic synaptic transmission in Ts65Dn neocortex.** (A–C) Global normalization of *DSCAM* gene dosage in Ts65Dn mice. mIPSCs were recorded from brain slices from euploid, Ts65Dn, and Ts65Dn:*DSCAM*+/+/− mice, which were obtained by mating female Ts65Dn and male *DSCAM^2j* mice. (**A**) Representative traces of mIPSCs from PyNs in layer II/III of the ACC. (**B, C**) Quantification of mIPSC frequency (**B**) and amplitude (**C**). A total of 6 euploid control, 6 Ts65Dn, and 6 Ts65Dn:*DSCAM*+/+/− mice were analyzed. For each mouse, 2–4 PyNs were recorded. (**D-F**) Normalization of *DSCAM* gene dosage in GABAergic neurons in Ts65Dn mice. *Lhx6*-Cre mice were crossed with *DSCAM^flox* mice to generate *Lhx6*-Cre:*DSCAM^flox* mice in the Ts65Dn background (Ts65Dn:*Lhx6*-Cre:*DSCAM^flox*). (**D**) Representative traces of mIPSCs from PyNs in layer II/III of the ACC in euploid:*Lhx6*-Cre, Ts65Dn:*Lhx6*-Cre, and Ts65Dn:*Lhx6*-Cre:*DSCAM^flox* brain slices. (**E, F**) Quantification of mIPSC frequency (**E**) and amplitude (**F**). A total of 4 euploid control, 4 Ts65Dn, and 4 Ts65Dn:*Lhx6*-Cre: *DSCAM^flox* mice were analyzed. For each mouse, 2–4 PyNs were recorded. One-way ANOVA for multigroup comparisons and post hoc Student *t* tests for pair-wise comparisons. **: $p < 0.01$; ***: $p < 0.001$; ns: not significant ($p > 0.05$). The data underlying this Figure can be found in https:// doi.org/10.5281/zenodo.7714234. ACC, anterior cingulate cortex; DSCAM, Down syndrome cell adhesion molecule; mIPSC, miniature inhibitory postsynaptic current; PyN, pyramidal neuron.

2E). The amplitude of mIPSC was not different between euploid and Ts65Dn mice or between Ts65Dn and Ts65Dn:*Lhx6*-Cre:*DSCAM^flox* (Fig 2D and 2F). These results are consistent with the morphological results showing that the extra copy of *DSCAM* gene leads to an increase in the number of GABAergic boutons on PyN somas (Fig 1C–1F).

Basket and ChC cells constitute the parvalbumin-expressing (PV+) interneurons in neocortex [40]. Notably, normalizing DSCAM levels did not rescue the increased density of PV

+ neurons in the ACC region of Ts65Dn mice (S5A and S5B Fig). Thus, the overexpressed DSCAM in the Ts65Dn neocortex causes excessive GABAergic synaptic transmission by increasing the number of GABAergic boutons and not by affecting the number of PV + GABAergic neurons.

## DSCAM overexpression increases ChC presynaptic terminals and boutons in Ts65Dn mice

Previous studies in *Drosophila* have demonstrated that Dscam levels determine the size of presynaptic terminals and that increased levels of Dscam lead to excessive growth of presynaptic terminals [3,49]. Whether an increase in DSCAM also causes overgrowth of presynaptic terminals in mice is unknown. Compared to basket cells [50], the morphology of neocortical ChCs is relatively stereotypical and reliably quantifiable [51–54]. Each ChC innervates roughly 200 PyNs at their axon initial segments (AISs) [55], where action potentials are generated [56,57].

We used the Nkx2.1-CreERT2 mouse line and the tdTomato reporter line Ai14 to label single ChCs in the neocortex [58] (Fig 3A). A single ChC extends only a few dendritic branches but several hundreds of presynaptic terminals, called axonal cartridges, each of which innervates an AIS of a PyNs [59]. The axonal cartridges and presynaptic boutons of single ChCs were quantified double-blindly at P28, a time after ChC development is complete [59]. The ChCs in layers II/III of the ACC were analyzed because the morphology of ChCs in this region is most stereotypical.

To determine whether the morphology of ChC axon terminals is altered in Ts65Dn neocortex and whether *DSCAM* triplication contributes to any morphological changes, we integrated Nkx2.1-CreERT2 and Ai14 in Ts65Dn and crossed female Ts65Dn mice with male *DSCAM* +/− mice. This genetic scheme yielded regular trisomic Ts65Dn mice (Ts65Dn:*DSCAM*+/+/+) as well as Ts65Dn that contained only 2 functional copies of *DSCAM* (Ts65Dn:*DSCAM*+/+/−) (Fig 3A).

Compared to euploid littermates, the total cartridge length, which is the sum of individual cartridges in the quantified volume, was increased by 42% (Fig 3B and 3C). In addition, ChCs in Ts65Dn mice showed a significant increase in both the number and size of synaptic boutons. The average bouton number per ChC was increased by 36% (Fig 3D and 3E), and the average size of presynaptic boutons was enlarged by 22% (Fig 3D and 3F). The average interbouton distance between neighboring boutons and the AIS length were not affected in the trisomy mice (S7A and S7B Fig). Thus, we found significantly increased presynaptic terminals and boutons in single ChCs in Ts65Dn mice.

Normalizing DSCAM levels rescued ChC presynaptic overgrowth in Ts65Dn mice. Compared with Ts65Dn littermates, the total cartridge lengths of single ChCs were reversed to levels indistinguishable from euploid in Ts65Dn:*DSCAM*+/+/− mice (Fig 3B and 3C). In addition, the increased bouton number and bouton size were rescued by normalizing DSCAM expression (Fig 3D–3F). No change was observed in the interbouton distance or the AIS length, as compared to either euploids or Ts65Dn mice (S7A and S7B Fig). These results demonstrate that ChC presynaptic terminal overgrowth in Ts65Dn mice is mainly caused by DSCAM overexpression.

## Loss of *DSCAM* reduces the number of GABAergic boutons on PyN soma and axon initial segments

The studies presented above show that DSCAM overexpression in the trisomy mouse model leads to excessive GABAergic boutons on PyN somas and AIS. Does loss of *DSCAM* lead to the converse phenotypes? To answer this question, we evaluated perisomatic GABAergic

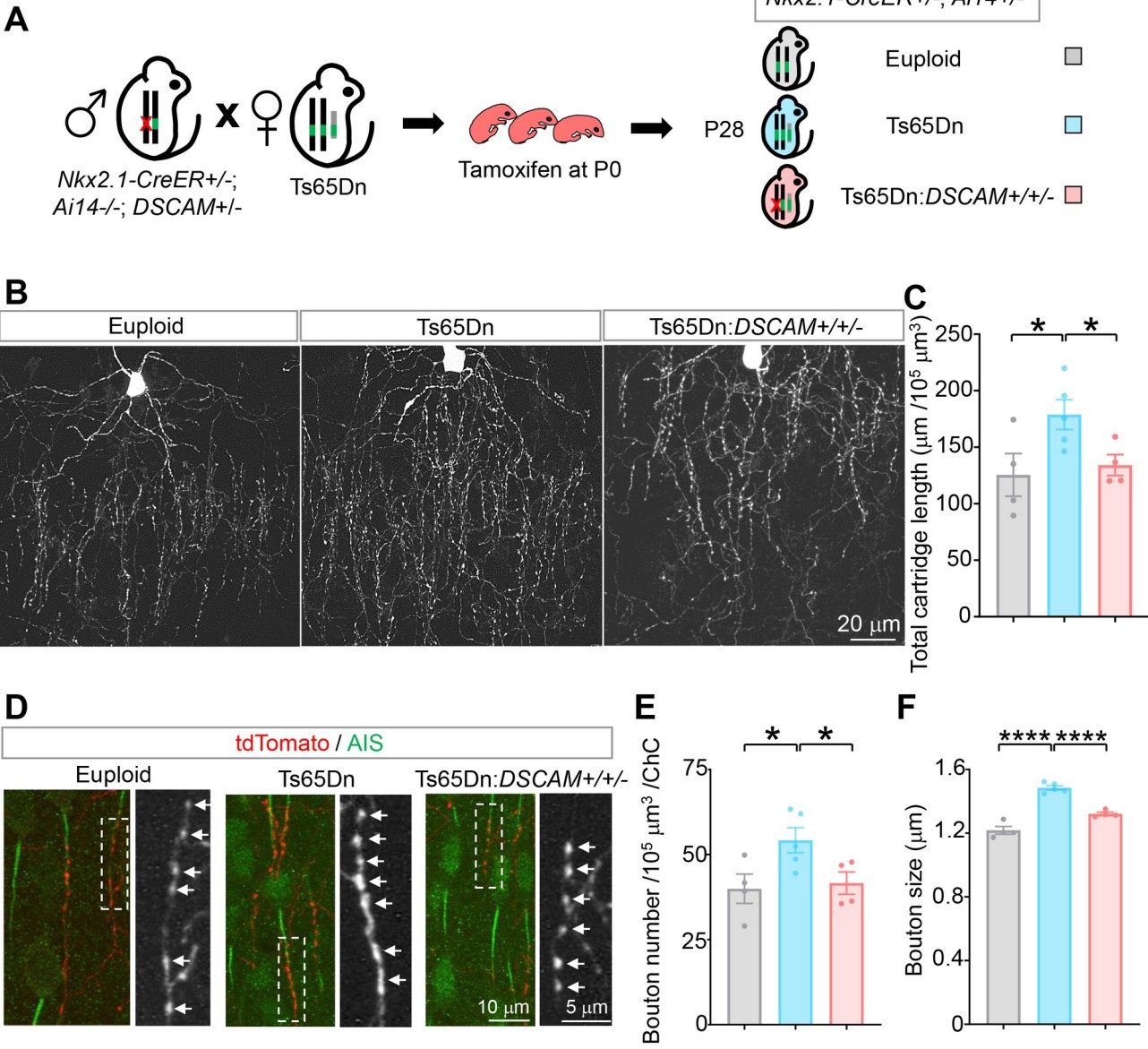

**Fig 3. Normalizing DSCAM expression rescues the overgrowth of ChC axon cartridges and presynaptic boutons in Ts65Dn mice.** (**A**) A schematic of the procedure that produced the mice for the experiments. (**B**) Representative images of single ChCs in layer II/III of the ACC. Shown are ChCs of euploid, Ts65Dn, and Ts65Dn with *DSCAM* allele normalized (Ts65Dn:*DSCAM+/+/−*) mice at P28. Scale bar, 20 μm. (**C**) Quantification of the total cartridge length. For each ChC, all axon cartridges (approximately 15–40) innervating the AIS of PyNs in a volume of 120 μm (length) × 80 μm (width) × 30 μm (thickness) with the ChC cell body in the top middle were analyzed (S6 Fig). For each mouse, 4–6 ChCs were analyzed; 4 euploid, 5 Ts65Dn, and 4 Ts65Dn:*DSCAM+/+/−* mice were analyzed. (**D**) Representative images of ChC axon cartridges innervating the AIS of PyNs. Cartridges of single ChCs were labeled with tdTomato (Red). The AIS of PyNs were labeled by anti-phospho-IκB (pIκB, green). The arrows point to presynaptic boutons of ChCs in the boxed regions. (**E, F**) Quantification of bouton number (**E**) and size (**F**). For each ChC, all boutons in axon cartridges that innervate AIS in the defined volume (S6 Fig) were analyzed for bouton numbers; boutons in the 10 cartridges nearest to the cell body were analyzed for bouton sizes. For each mouse, 4–6 ChCs were analyzed, and 4 euploid, 5 Ts65Dn, and 4 Ts65Dn:*DSCAM+/+/−* mice were analyzed. Each dot in the charts represent 1 mouse. One-way ANOVA for multigroup comparisons and post hoc Student *t* tests for pair-wise comparisons. *: $p < 0.05$; ****: $p < 0.0001$. The data underlying this Figure can be found in https://doi.org/10.5281/zenodo.7714234. ACC, anterior cingulate cortex; AIS, axon initial segment; ChC, chandelier cell; DSCAM, Down syndrome cell adhesion molecule; PyN, pyramidal neuron; P28, postnatal day 28.

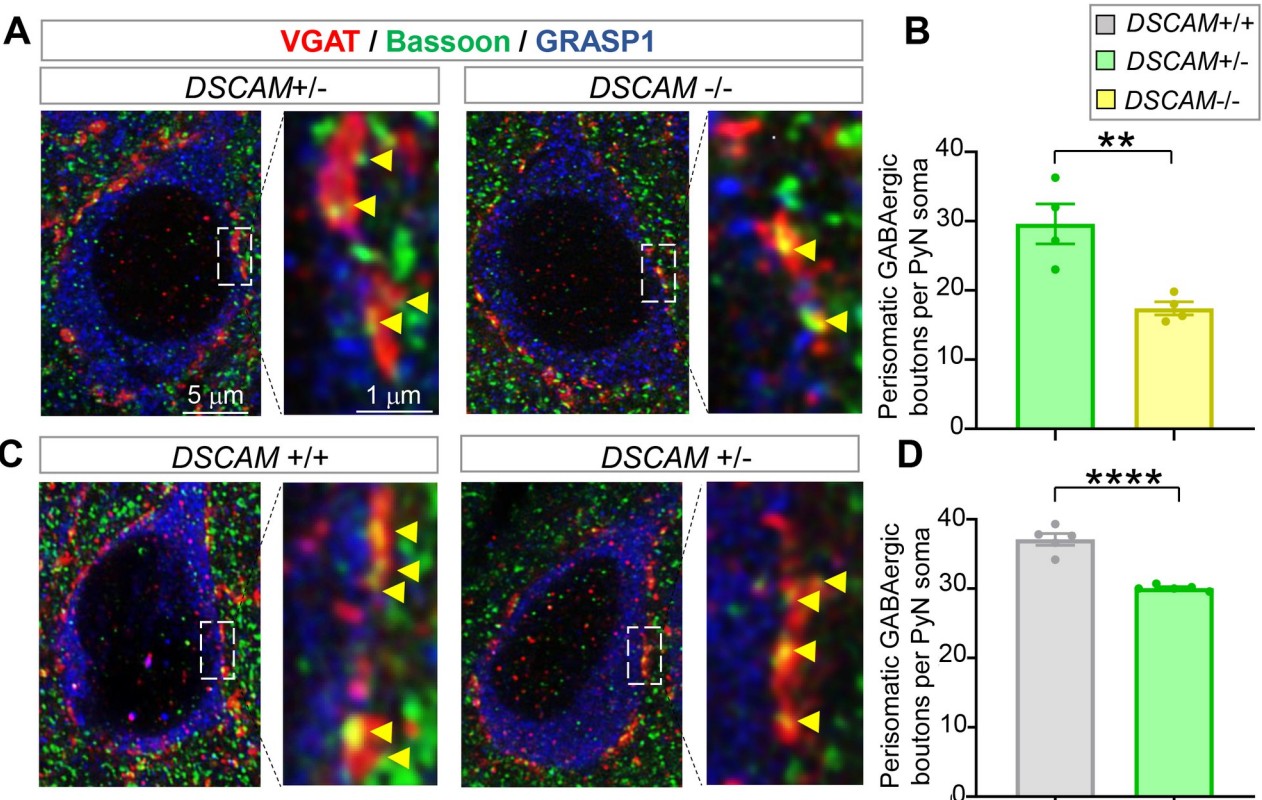

**Fig 4. Loss of *DSCAM* impairs the perisomatic GABAergic boutons innervating PyNs in layer II/III of the ACC.** (**A** and **C**) Representative images of perisomatic GABAergic boutons innervating PyNs of $DSCAM^{2j/+}$ (+/−) and $DSCAM^{2j/2j}$ (−/−) mice (**A**) as well as those of $DSCAM^{+/+}$ (+/+) and $DSCAM^{2j/+}$ (+/−) mice (**C**). (**B** and **D**) Quantifications. Each data point in the chart represents the mean in 1 mouse. Student *t* test. **: $p < 0.01$; ****: $p < 0.0001$. The data underlying this Figure can be found in https://doi.org/10.5281/zenodo.7714234. ACC, anterior cingulate cortex; DSCAM, Down syndrome cell adhesion molecule; PyN, pyramidal neuron.

boutons in the loss-of-function *DSCAM* mutant $DSCAM^{2j}$. As reported previously [37,38], DSCAM protein was not detected in $DSCAM^{2j/2j}$ ($DSCAM^{−/−}$) by western blotting (S8A Fig). Brain sections were triply stained with anti-Bassoon (for presynaptic active zone), anti-VGAT (for presynaptic GABAergic boutons), and anti-GRASP1 (for PyN soma and proximal dendrites). We quantified the data in a double-blinded fashion to avoid experimenter bias and compared the $DSCAM^{−/−}$, $DSCAM^{+/−}$, and $DSCAM^{+/+}$ groups. The average number of GABAergic boutons around each PyN soma was reduced 41% in the neocortices of $DSCAM^{−/−}$ mice compared to $DSCAM^{+/−}$ mice (Fig 4A and 4B). The level of DSCAM protein in $DSCAM^{+/−}$ mice was 65% of that in $DSCAM^{+/+}$ mice (S8B Fig). Consistent with the notion that DSCAM function in synaptic development is dose dependent [3], PyNs in $DSCAM^{+/−}$ mice also had fewer perisomatic GABAergic boutons than those in $DSCAM^{+/+}$ mice (Fig 4C and 4D). These data suggest that *DSCAM* is required for forming the proper number of GABAergic boutons on PyN somas in the neocortex. Moreover, the dose dependence of DSCAM function highlights the importance of DSCAM expression levels in regulating synaptic development.

Similar to its role in presynaptic development in *Drosophila* [3,49], loss of *DSCAM* significantly impeded the development of ChC presynaptic terminals. Compared to heterozygous littermates, the average total cartridge length was reduced by 24% (Fig 5A and 5B). Moreover, the average numbers and sizes of presynaptic boutons were significantly reduced by 20% and

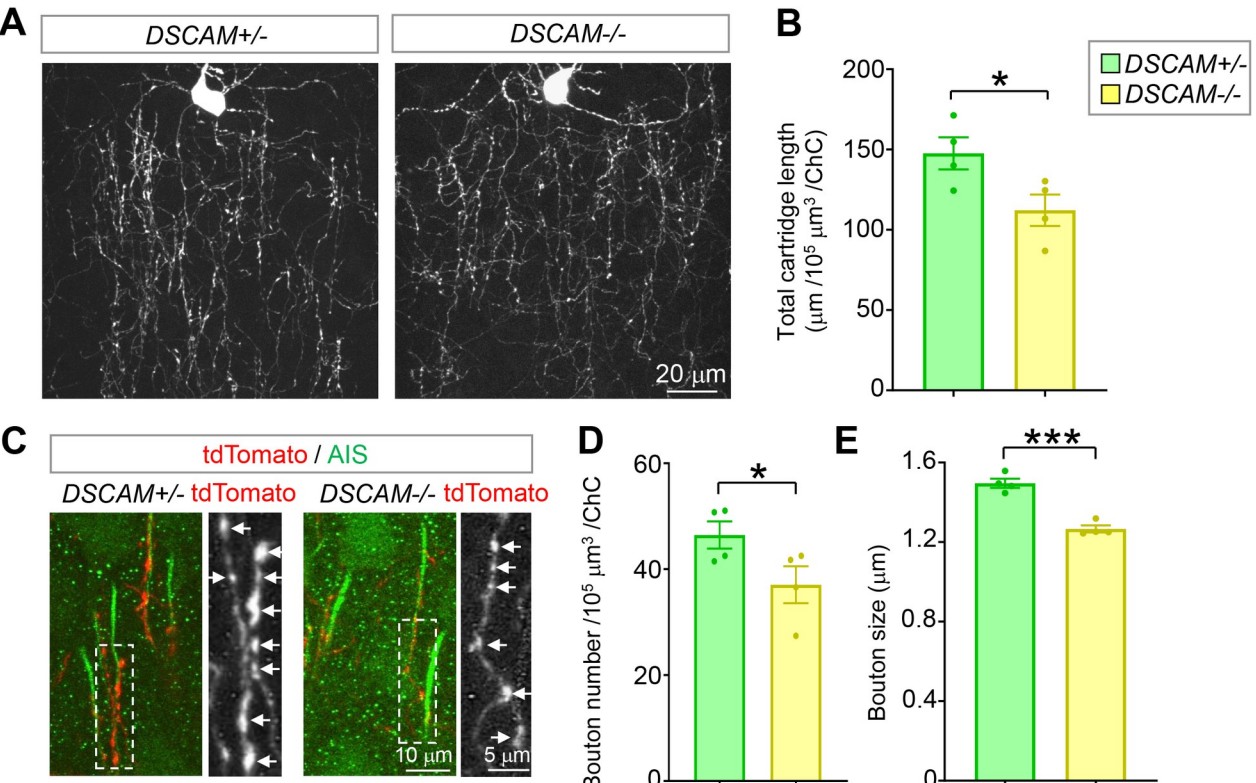

**Fig 5. Loss of *DSCAM* impairs the growth of ChC axon cartridges and boutons.** (**A**) Representative images of single ChCs in layer II/III of ACC. Shown are ChCs of *DSCAM²ʲ/⁺* (+/−) and *DSCAM²ʲ/²ʲ* (−/−) mice at P28. All ChC images in this paper are from this brain region of P28 mice. Scale bar, 20 μm. (**B**) Quantification of the total cartridge length (A). For each ChC, all axon cartridges (approximately 15–40) innervating the AIS of PyNs in a volume of 120 μm (length) × 80 μm (width) × 30 μm (thickness) with the ChC cell body in the top middle were analyzed (S6 Fig). For each mouse, 4–6 ChCs were analyzed, and 4 *DSCAM+/−* and 4 *DSCAM−/−* mice were analyzed. (**C**) Representative images of ChC axon cartridges innervating the AIS of PyNs. Cartridges of single ChCs were labeled with tdTomato (Red). The AIS of PyNs were labeled by anti-phospho-IκB (pIκB, green). The arrows point to presynaptic boutons of ChCs in the boxed regions. (**D**, **E**) Quantification of bouton number (**D**) and size (**E**). For each ChC, all boutons in axon cartridges that innervate AIS in the defined volume (S6 Fig) were analyzed for bouton numbers; boutons in the 10 cartridges nearest to the cell body were analyzed for bouton sizes. In each mouse, 4–6 ChCs were analyzed. A total of 4 *DSCAM+/−* and 4 *DSCAM −/−* mice were analyzed. Each dot represents 1 mouse. Student *t* test *: $p < 0.05$; ***: $p < 0.001$. The data underlying this Figure can be found in https://doi.org/10.5281/zenodo.7714234. ACC, anterior cingulate cortex; AIS, axon initial segment; ChC, chandelier cell; DSCAM, Down syndrome cell adhesion molecule; PyN, pyramidal neuron; P28, postnatal day 28.

15%, respectively (Fig 5C–5E). No difference was observed in the interbouton distance or AIS length between these 2 groups (S7C and S7D Fig), indicating that *DSCAM* does not regulate bouton density. These results demonstrate that *DSCAM* is required for ChC presynaptic development in mice.

Consistent with the observation that normalizing DSCAM levels did not rescue the increased PV+ neuron density in Ts65Dn mice (S5A and S5B Fig), the number of PV+ neurons was unaffected by loss of *DSCAM* (S5C and S5D Fig).

## Loss of *DSCAM* impairs GABAergic synaptic transmission to PyNs

To determine whether defective GABAergic bouton numbers caused by loss of *DSCAM* impairs GABAergic synaptic transmission, patch-clamp recordings in the whole-cell configuration were employed to record inhibitory currents in the PyNs in layers II/III of the ACC from acute neocortical brain slices. Consistent with impaired axonal growth and bouton number in ChCs, we found that the average frequency of mIPSCs was approximately 35% less in

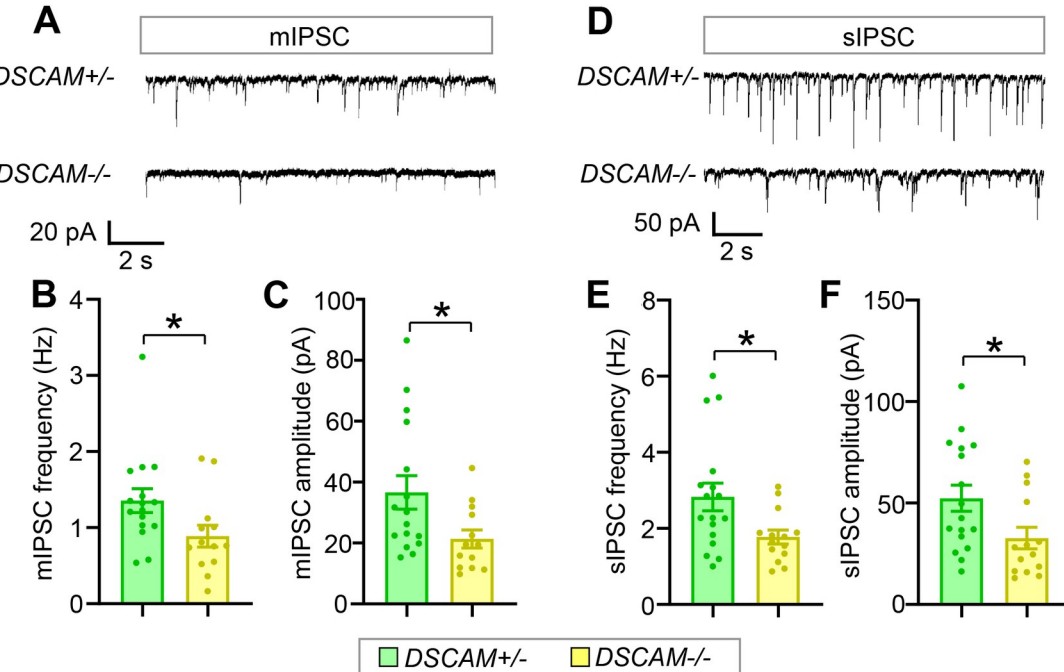

**Fig 6. Loss of *DSCAM* impairs GABAergic synaptic transmission to neocortical PyNs.** (**A**) Representative traces of mIPSCs from PyNs in layer II/III of the ACC in *DSCAM*+/− and *DSCAM*−/− brain slices. (**B, C**) Quantification of mIPSC frequency (**B**) and amplitude (**C**). For each mouse, 3–5 PyNs were recorded. A total of 5 *DSCAM*+/− and 4 *DSCAM*−/− mice were analyzed. N: 16 for *DSCAM*+/−, and 13 for *DSCAM*−/−. (**D**) Representative traces of sIPSCs from PyNs in layer II/III of ACC in *DSCAM* +/− and *DSCAM*−/− brain slices. (**E, F**) Quantification of sIPSC frequency (**E**) and amplitude (**F**). Approximately 3–5 PyNs were recorded for each mouse. A total of 5 *DSCAM*+/− and 4 *DSCAM*−/− mice were analyzed. N: 17 for *DSCAM*+/−, 14 for *DSCAM*−/−. Student *t* test. *: $p < 0.05$. The data underlying this Figure can be found in https://doi.org/10.5281/zenodo.7714234. ACC, anterior cingulate cortex; DSCAM, Down syndrome cell adhesion molecule; mIPSC, miniature inhibitory postsynaptic current; PyN, pyramidal neuron; sIPSC, spontaneous inhibitory postsynaptic current.

*DSCAM*$^{-/-}$ mice than that in heterozygous littermates (Fig 6A and 6B). In addition, the average amplitude of mIPSCs was reduced by 42%, suggesting that postsynaptic responses were impaired by loss of *DSCAM* (Fig 6C). Similar changes were observed in the frequency and amplitude of sIPSCs (Fig 6D–6F).

## ChC axon terminal growth and bouton number are positively coupled to neocortical DSCAM levels

Presynaptic terminal growth and synaptogenesis are concurrent processes for forming proper neuronal connections [60]. Little is known about whether and how these 2 events are orchestrated in mammalian GABAergic interneurons. The sparse labeling of ChCs offers an opportunity to address these questions at single-cell resolution. We examined the relationship between cartridge length and 3 morphological aspects of synaptic boutons, namely bouton number, size, and density (as reflected by the interbouton distance), in single ChCs. We found a strong correlation between the cartridge length and the bouton number of each ChC in both wild-type (euploid) ($R^2 = 0.82$, $p < 10^{-7}$) and Ts65Dn mice ($R^2 = 0.79$, $p < 10^{-15}$) (S9A Fig). Although DSCAM positively regulated both cartridge length and bouton number (Figs 3B–3F and 5), loss of *DSCAM* did not impair their coupling ($R^2 = 0.89$, $p < 10^{-8}$) (S9B Fig). There was also a significant correlation between cartridge length and bouton size in both wild-type and Ts65Dn mice (S9C Fig), though it was weaker than that between cartridge length and bouton number. Loss of *DSCAM* seemed to mildly impair the coupling between cartridge length

and bouton size (S9D Fig). There was no correlation between cartridge length and bouton density in any of the genotypes tested (S9E and S9F Fig). Taken together, these results suggest that presynaptic terminal growth and bouton number are positively coupled in ChC development.

In *Drosophila*, Dscam levels determine the size of presynaptic arbors [3]. Western blotting results suggested that DSCAM levels in the neocortex exhibited individual variations in euploid, Ts65Dn, and Ts65Dn:*DSCAM*+/+/− mice (S10A Fig). We plotted the neocortical DSCAM level of each mouse, as assayed by western blotting, against the average cartridge length, bouton number, size, or density in each mouse. We observed a strong correlation between DSCAM levels and cartridge lengths, bouton number, and bouton sizes in mice (S10B–S10D Fig). In contrast, no correlation was found between DSCAM levels and interbouton distance (S10E Fig), again supporting that bouton density is not regulated by DSCAM (S7 Fig). The dose dependence highlights the importance of DSCAM expression levels in regulating ChC presynaptic development.

## Discussion

In this study, we show that the extra copy of *DSCAM* in a mouse model of DS leads to presynaptic overgrowth in ChCs and basket cells, 2 major types of inhibitory neurons in the neocortex. The hyper-innervation and excessive GABAergic inhibition of PyNs in Ts65Dn were rescued by normalizing the DSCAM levels. The converse phenotypes were observed in *DSCAM* loss-of-function mutant mice. The sensitivity of GABAergic synapse development and function to DSCAM expression levels suggest that dysregulated DSCAM expression may underlie GABAergic dysfunction in neurological disorders that exhibit abnormal DSCAM expression, including DS, ASDs, intractable epilepsy, bipolar disorders, and, possibly, Fragile X syndrome.

The effects of DSCAM are unlikely to stem from mitotic proliferation for several reasons. First, DSCAM is expressed in differentiating neurons but not mitotic progenitors [1,61–63]. Second, our results show that normalizing DSCAM levels does not rescue the increased density of PV+ neurons in the ACC region of Ts65Dn mice (S5A and S5B Fig). Consistently, the number of cortical PV+ neurons was not affected in *DSCAM* null mice (S5C and S5D Fig).

### DSCAM expression levels determine presynaptic terminal sizes in mammalian neurons

The stereotypical morphology of ChCs axon arbors and presynaptic terminals are advantageous for quantitative assessment [51,52]. Moreover, recent advances in genetic labeling of ChCs have allowed sparse labeling of single ChCs for quantifying presynaptic terminals at single-cell resolution [58,64]. By taking advantage of this system, we show in this study that DSCAM overexpression causes presynaptic overgrowth of ChCs in the neocortex. Notably, ChC cartridge length, bouton number, and bouton sizes correlate with DSCAM expression levels (S10B–S10D Fig), suggesting the sensitivity of ChC development to DSCAM levels. Thus, this work supports a conserved role of *DSCAM* in regulating presynaptic terminal growth in *Drosophila* and mice.

Whether *DSCAM* functions cell-autonomously to promote ChC presynaptic terminal growth remains an open question due to technical difficulties. Genetic deletion of *DSCAM* in single ChCs is challenging because the Cre activity in Nkx2.1-CreER mouse line is weak. Since administration of tamoxifen cannot guarantee the deletion of target genes in all or vast majority of cells in floxed mice, immunostaining or in situ hybridization is required to confirm the deletion. However, available anti-DSCAM antibodies are not conducive for immunostaining

in the neocortex [6]. Moreover, the experimental procedures for in situ hybridization are not compatible with morphological studies of ChCs by immunostaining. Future studies with reliable ChC-specific genetic deletions (e.g., via a stronger Cre) will determine whether *DSCAM* functions cell-autonomously.

Our studies using an *Lhx6*-Cre mouse line that targets GABAergic interneurons show that the extra copy of *DSCAM* in GABAergic neurons leads to the excessive GABAergic boutons on PyN somas (Fig 1D and 1F) and increased mIPSC frequencies in PyNs (Fig 2D–2F). The *Lhx6*-Cre transgene is not only expressed in basket cells, which forms the perisomatic GABAergic synapses on PyNs [40,41], but also expressed in ChC cells and somatostatin + GABAergic neurons [46,47]. As such, these results do not demonstrate that the extra copy of *DSCAM* in basket cells causes the excessive boutons or increased mIPSC frequencies. Testing the cell-autonomous functions of *DSCAM* in perisomatic synapse development requires removing the extra copy of *DSCAM* specifically in basket cells.

## The role of DSCAM in GABAergic synaptic transmission in the neocortex

Whole-cell patch-clamp recordings showed that mIPSC frequency of PyNs was increased in Ts65Dn mice and that normalizing DSCAM levels rescued this change (Fig 2). Moreover, mIPSC frequency of PyNs was reduced in $DSCAM^{-/-}$ mice (Fig 6A and 6B). Because mIPSC frequency indicates input numbers and presynaptic release sites, these findings are consistent with a role of DSCAM in increasing bouton numbers in basket cells (Figs 1C, 1E, and 4).

We found no difference in mIPSC amplitude of PyNs between euploid and Ts65Dn mice (Fig 2A and 2C), suggesting that postsynaptic responses may not be affected by the trisomy. By contrast, mIPSC amplitude of PyNs was reduced in $DSCAM^{-/-}$ mice (Fig 6C), suggesting that postsynaptic responses were impaired by loss of *DSCAM*. Taken together, these results indicate that while DSCAM might be essential for postsynaptic responses at the inhibitory synapses formed on PyNs, their overexpression is insufficient to change the postsynaptic responses, at least in Ts65Dn mice.

## A potential role of DSCAM in excitatory synaptic transmission in the neocortex in DS mouse model

In this study, we found that the sEPSC frequency was also increased in PyNs in the ACC, which was reduced by normalizing *DSCAM* gene dosage (S4I Fig). This result suggests that the activity of excitatory presynaptic terminals is also regulated by DSCAM expression levels. By contrast, we did not observe any difference in the frequency of sEPSCs in ChCs (S11 Fig), suggesting that HSA21 trisomy has cell type–specific effects on excitatory synaptic transmission in the neocortex. A complete analysis of DSCAM's contribution to the excitatory synaptic transmission in DS models is an interesting future direction.

Similar to PyNs, ChCs did not exhibit any difference in either their single action potential or membrane properties among euploid, Ts65Dn, and Ts65Dn:*DSCAM*+/+/− mice (S11A–S11F Fig), except for a minor change in the threshold in Ts65Dn mice (S11B Fig). This minor change has no impact on neuronal excitability since rheobase was not affected (rheobase is defined as the minimal current required to evoke an action potential) (S11H Fig). Different from PyNs, both ChCs in Ts65Dn mice and those in Ts65Dn:*DSCAM*+/+/− mice exhibited lower firing frequencies than in euploid mice (S11G Fig). This suggests that ChCs might receive increased inhibitory inputs or reduced excitatory inputs, independent of *DSCAM*. The fact that PyN firing rates are indistinguishable in euploid, Ts65Dn, and Ts65Dn:*DSCAM* +/+/− mice (S4G Fig) suggests the excessive GABAergic inhibition on PyNs might be balanced by excessive glutamatergic excitation.

## The coupling between presynaptic terminal growth and bouton number in ChCs

Among the different types of GABAergic neurons, ChCs are thought to be a powerful source of inhibition in the neocortex [65]. These neurons form unique axo-axonic GABAergic synapses that selectively innervate PyNs at their AIS, where action potentials are generated [56,57]; each ChC innervates roughly 200 PyNs [55]. Impaired ChC presynaptic growth is present in individuals with epilepsy and schizophrenia, both of which are thought to be caused, in part, by disrupted GABAergic signaling [65–69]. Recent studies demonstrate that deleting *Erbb4*, a schizophrenia-associated gene, in ChCs causes a schizophrenia-like phenotype in mice, indicating a causal relationship between ChC defects and schizophrenia [46,54]. In addition to ErbB4, Neuregulin 1, DOCK7, Fgf13, and L1CAM have also been found to regulate synapse formation between ChCs and PyNs [51–53,70].

In the present study, detailed investigation of the cartridge growth and bouton numbers at single-cell resolution uncovered several interesting features of ChC development. First, presynaptic cartridge growth is strongly associated with bouton number and, to a lesser extent, bouton size (S9A–S9D Fig). This observation supports the synaptotropic model proposing that synaptogenesis stabilizes axonal arbor growth in neurodevelopment [71]. Second, the factors that regulate cartridge growth and synaptogenesis are not necessarily the factors coupling these 2 processes. Although DSCAM regulates both cartridge growth and bouton number, the coupling of these 2 processes remains intact in mice that are deficient of *DSCAM* function, suggesting that DSCAM is not the coupling factor. Identifying and distinguishing coupling factors from regulators is an important step toward a mechanistic understanding of ChC development.

## Possible mechanisms by which DSCAM regulates inhibitory synapses

By restoring DSCAM level in trisomy background, we demonstrate the causality of DSCAM's role in inhibitory synaptic changes in a DS mouse model. The detailed molecular mechanism related to DSCAM's role remains elusive. Conditional knockout of 1 copy of the *DSCAM* gene in GABAergic neurons in the trisomy background reduced the number of GABAergic boutons and mIPSC frequency to the normal level (Figs 1D, 1F, 2D–2F, and S3D–S3F), suggesting that increased level of DSCAM in GABAergic neurons is responsible for the synaptic changes. Moreover, we did not observe changes in the size of PyNs or the length of AIS (S1D, S7B, and S7D Figs), further supporting that DSCAM's role in inhibitory synapses is caused by their expression in GABAergic neurons. Previous studies have shown several molecules that mediate signaling downstream of DSCAM, including Abelson tyrosine kinase [49], Pak1 [72–74], and Dock [74]. Moreover, DSCAM might regulate electric activity of GABAergic neurons and consequently changes their presynaptic terminals and synapses. These are interesting possible mechanisms to investigate in the future.

## Insights into other brain disorders

Altered DSCAM expression levels have been associated with several brain disorders, including DS, ASD, intractable epilepsy and bipolar disorder [6–11], and, possibly, Fragile X syndrome [3,4,12,13]. The present work has established a causal relationship between dysregulated DSCAM levels and the developmental and functional defects of GABAergic neurons. The dose-dependent function of DSCAM suggests that dysregulated DSCAM levels may be a common pathogenic driver of GABAergic dysfunctions related to neurological diseases. For example, genetic analyses have revealed multiple disruptive single-nucleotide variants (SNVs) and

copy number variants (CNVs) in the *DSCAM* gene in idiopathic ASD individuals, raising the possibility of altered DSCAM expression levels [7–9,75]. Given the regulation of GABAergic signaling by DSCAM levels discovered in the present study and the established role of impaired GABAergic signaling in ASD [76], one may hypothesize that reduced DSCAM expression causes impaired GABAergic signaling in ASD. In support of this idea, some ASD individuals with CNV of deleted enhancer region in DSCAM show nonfebrile seizures [9], a symptom also found in mice with deficient DSCAM function [77]. It is thus important to examine DSCAM levels in postmortem samples of autistic individuals and determine whether altered DSCAM expression causes GABAergic dysfunctions in ASD mouse models.

## Methods and materials

### Ethics statement

The Institutional Animal Care and Use Committees at University of Michigan reviewed and approved all experimental procedures involving mice (protocol numbers PRO00005862 and PRO00007778). For immunohistochemistry, mice were killed by $CO_2$, immediately followed by intracardial infusion (see below). For electrophysiology, mice were decapitated under iso-flurane and USP anesthesia, and the brains were quickly removed from the skull for further processing (see below).

### Mouse breeding and tamoxifen administration

Age-matched littermates were used for all experiments. The Ts65Dn mouse line (Ts(17(16))65Dn) was purchased from Jackson Laboratory (Stock No: 005252) and maintained by crossing with C57BL/C3H F1 hybrid males created from breeding C57BL/6J (Stock No: 000664) with the C3Sn.BLiA-*Pde6b*+/DnJ line (Stock No: 003648).

For studying the perisomatic GABAergic boutons formed on PyNs, female Ts65Dn mice (Ts(17(16))65Dn) on a C57BL/C3H mixed background were bred to male C3H/HeDiSn-*Dscam*[2j]/GrsrJ mice from Jackson Laboratory (Stock No: 006038) to generate euploid, Ts65Dn, and Ts65Dn:*DSCAM*+/+/− littermates for data collection.

To determine whether the extra copy of *DSCAM* gene within GABAergic neurons leads to the excessive GABAergic boutons on PyN somas, we normalized *DSCAM* gene dosage within cortical GABAergic neurons. *Lhx6*-Cre mice were crossed with *DSCAM*[flox] mice (Stock No: 17689) to generate male *Lhx6*-Cre+/−, *DSCAM*[flox] +/− (*Lhx6*-Cre/*DSCAM*[flox]) mice, which, in turn, were bred to female Ts65Dn (Ts(17(16))65Dn; on a C57BL/C3H mixed background) to generate (1) *Lhx6*-Cre+/−:euploid; (2) Ts65Dn:*Lhx6*-Cre+/−; and (3) Ts65Dn:*Lhx6*-Cre +/−/*DSCAM*[flox] littermates. Tamoxifen (Sigma, T5648-1G), dissolved in corn oil, was delivered to pregnant mothers at E18 by intraperitoneal injection at the dosage of 80 mg/kg body weight to normalize *DSCAM* gene dosage within cortical GABAergic neurons.

For labeling single ChCs in Ts65Dn mice, Ts65Dn females were crossed with *Nkx2.1*-CreER+/−, Ai14−/−, *DSCAM*+/− male mice of mixed C57BL/C3H background to obtain euploid, Ts65Dn, and Ts65Dn:*DSCAM*+/+/− littermates with *Nkx2.1*-CreER+/−, Ai14 +/− transgenes. Tamoxifen (Sigma, T5648-1G), dissolved in corn oil, was delivered to P0 pups by intraperitoneal injection at the dosage of 80 mg/kg body weight. Pseudodams were prepared in advance to lactate the pups as the pups were often discarded by Ts65Dn dams after tamoxifen administration. For labeling single ChCs in *DSCAM*[2j] mice, female *DSCAM* +/− mice of C3H background were crossed with *Nkx2.1*-CreER+/−, Ai14−/−, *DSCAM* +/− male mice of mixed C57BL/C3H background to obtain *DSCAM*+/− and *DSCAM*−/− pups with *Nkx2.1*-CreER+/−, Ai14+/− transgenes. Tamoxifen was delivered to P0 pups as described above. Strong pups were killed at P0 to keep the litter size 4 to 5 to increase the surviving rate

of *DSCAM−/−* pups. *Nkx2.1*-CreER+/−, Ai14−/−, *DSCAM+/−* male mice were removed from the cage before female laboring as the male adults is likely to attack and kill *DSCAM−/−* pups.

Because *DSCAM−/−* pups are weak, in the study of perisomatic GABAergic boutons, we killed the wild-type littermates at P5 to P7 to promote the survival of *DSCAM−/−* pups. It was thus almost impossible to collect wild-type, heterozygotes, and homozygotes from the same litter. Therefore, we compared perisomatic GABAergic boutons between *DSCAM+/+* and *DSCAM+/−* mice and between *DSCAM+/−* and *DSCAM−/−* mice (Fig 4).

For electrophysiology recordings, Ts65Dn females were crossed with C3H *DSCAM +/−* male mice to obtain euploid, Ts65Dn, and Ts65Dn: *DSCAM+/+/−* littermates. C3H *DSCAM+/−* mice were interbred to obtain *DSCAM+/−* and *DSCAM−/−* pups for recordings.

PCR genotyping was performed on purified tail tips at least 2 times to confirm the genotype according to the protocol by Jackson Laboratory. All work involving mice was approved by the University of Michigan Institutional Animal Care and Use Committee (IACUC) and in accordance with the National Research Council Guide for the Care and Use of Laboratory Animals (NIH).

### Immunohistochemistry

P28 mice were killed by $CO_2$ and immediately followed by intracardial infusion of 4% paraformaldehyde (PFA) with 4% sucrose in 1× PBS. The brains were then removed and postfixed in 4% PFA overnight at 4˚C, followed by incubation in 30% sucrose (wt/vol) for at least 1 day. The brains were embedded in the OCT compound (Fisher HealthCare), frozen in −20˚C overnight, and then sectioned to 100 μm-thick slices with a Leica CM3050S cryostat. Sectioned brain slices were kept in 1× PBS containing 0.05% sodium azide at 4˚C until immunostaining.

For ChC and AIS staining, coronal brain sections (100 μm) were blocked with 8% BSA in PBST (1× PBS + 0.1% Triton x-100) containing 0.05% sodium azide at 37˚C for 1 hour and then incubated with the following primary antibodies in the blocking solution at 37˚C overnight. The AIS was immunolabeled using an anti-phospho-IκB antibody (pIκB) [51,52,58,78], which strongly correlates with the AIS labeled by ankyrin-G antibodies in the mouse neocortex [79] (S6B and S6C Fig). Primary antibodies used were anti-mCherry (SICGEN, AB0081; 1:300) for neuronal morphology and anti-phospho-Iκβ-α (Cell Signaling, 14D4 rabbit monoclonal antibody; 1:500) for labeling the AIS. After washing 3 times (1 hour each time) at 37˚C in PBST, brain slices were incubated with the following secondary antibodies in the blocking solution at 37˚C overnight: donkey anti-goat-rhodamine RX (RRX) (Jackson ImmunoResearch, 705-297-003; 1:300), donkey anti-rabbit-Alexa Fluor 488 (Jackson ImmunoResearch, 711-545-152; 1:300). After washing 3× for a total of 1 hour at 37˚C in PBST, the slices were mounted in sRIMS [80].

The procedure for staining of GABAergic neurons and PyNs was the same as above, except that antibody incubations were done at room temperature (RT) and different primary antibodies were used. Primary antibodies used were anti-parvalbumin (Swant, PVG213; 1:1,000), anti-CAMKIIα (LSBio, 6G9; 1:5,000). Secondary antibodies used were donkey anti-goat-RRX (Jackson ImmunoResearch, 705-297-003; 1:500) and donkey anti-mouse-Alexa Fluor 488 (Jackson ImmunoResearch, 715-545-150; 1:500).

For staining perisomatic GABAergic boutons, antigen retrieval was conducted in 10 μM sodium citrate for 20 minutes at 95˚C [81]. After a brief rinse in PBS, the slices were blocked in blocking buffer (1× PBS, 0.3% Triton x-100, 3% normal donkey serum, 0.05% sodium azide) for 1 hour at RT and then incubated with the following primary antibodies in the blocking solution at 4˚C overnight: mouse anti-Bassoon (ENZO, SAP7F407; 1:1,000), guinea pig anti-VGAT (Synaptic System, 131 004; 1:1,000), rabbit anti-GRASP1 (1:2,000) [45]. After

washing 3 times (20 minutes each time) at RT in PBST, brain slices were incubated with the following secondary antibodies in the blocking solution at RT 3 hours: donkey anti-mouse-Alexa Fluor 488 (Jackson ImmunoResearch, 715-545-150; 1:300), donkey anti-guinea-pig-RRX (Jackson ImmunoResearch, 706-295-148; 1:300), donkey anti-rabbit-Alexa Fluor 647 (Jackson ImmunoResearch, 711-605-152; 1:300). The slices were mounted for imaging after washing 3 times for a total of 1 hour at RT in PBST.

## Image acquisition and quantitative analysis

All images were acquired from cortical layer II/III by using a Leica SP5 confocal microscope equipped with a resonant scanner, except the samples for perisomatic GABAergic boutons (see below). For ChCs imaging, different fluorescence channels were imaged sequentially with the pinhole set at airy 1. A 63× objective lens with a numerical aperture of 1.4 was used. Confocal image stacks were collected with 100 continuous optical sections at 0.3-μm z-steps. The cell body was positioned around the middle of the 30-μm in depth. The axonal cartridges and presynaptic boutons of single ChCs were quantified in a three-dimensional volume defined based on the position of the ChC soma (S6A Fig). Before quantification, the image stacks were maximally projected along the z-axis. A region of 120 μm (length) × 80 μm (width) with the cell body in the top middle was set for quantification. Cartridges and boutons that colocalized with AIS were quantified by the NeuroLucida software (MBF Bioscience). Cartridge/bouton number was defined as the number of cartridges/boutons within this region. AIS-colocalized cartridges and presynaptic boutons were quantified. Cartridge length was defined as the distance from the first bouton that colocalized with AIS to the last one colocalized with AIS in that cartridge. Bouton size was defined as the length of the bouton in parallel to the AIS. We quantified the sizes of boutons in the 10 cartridges nearest to the cell body. Interbouton distance was defined as the distance between 2 neighboring boutons.

Samples for perisomatic GABAergic boutons were imaged on a Leica SP8 confocal microscope with a 63× objective lens (NA 1.4). For each PyN, a single confocal image was taken at the z-position where PyN cell body occupied the most area. Images were deconvoluted with the Huygens software (Scientific Volume Imaging). Perisomatic VGAT+ and Bassoon+ puncta were quantified in defined PyN soma regions, as shown in S1C Fig, by the NeuroLucida software (MBF Bioscience). Perisomatic GABAergic boutons were counted as puncta that contained pixels positive for both VGAT and Bassoon.

To eliminate experimenter bias, these experiments were carried out in double-blind fashion. Two double-blind methods were used. First, the images acquired by the primary experimenter was coded and randomized by the second lab member. After the primary experimenter quantified the data, the data were decoded for statistical analysis. Second, the mouse brains from the primary experimenter were coded and randomized by the second lab member. The second lab member sent the coded mouse brains to the third lab member for sectioning. After the primary experimenter immunostained and quantified the encoded brain sections, the data were decoded for a final statistical analysis.

## Western blotting

Mouse neocortices were removed immediately after PFA perfusion. Similar volumes of tissues were homogenized in SDS sample buffer and loaded for electrophoresis. The proteins were transferred to PVDF membranes and blocked with 5% milk (wt/vol) for 1 hour at RT. The blots were incubated overnight at 4°C with goat anti-DSCAM (R&D Systems, AF3666; 1:500), anti-APP (Cell Signaling Technology, 2452; 1:1,000), or mouse anti-tubulin (12G10, 1:5,000, DSHB). After washing, blots were incubated for 2 hours with HRP-conjugated secondary

antibodies (1:5,000) and developed by chemiluminescence (Pierce, 32106). Quantification was performed using ImageJ.

## Electrophysiology recordings and analysis

Electrophysiological recordings of single action potential properties, firing frequency, and spontaneous and miniature postsynaptic currents were performed as described previously [81]. Brains were obtained from euploid, Ts65Dn, and Ts65Dn/*DSCAM*+/+/− mice at around P28. The animals were killed by decapitation under isoflurane and USP anesthesia. The brain was quickly removed from the skull and placed in 4˚C slicing solution containing (in mM) 62.5 NaCl, 2.5 KCl, 1.25 $KH_2PO4$, 26 $NaHCO_3$, 5 $MgCl_2$, 0.5 $CaCl_2$, 20 glucose, and 100 sucrose (pH maintained at 7.4 by saturation with 95% $O_2$ + 5% $CO_2$). Acute coronal brain slices (300 to 350 μm thick) containing layers II/III of the ACC neocortex were cut with a microtome (VF-300, Compresstome). The slices were then transferred to a holding chamber and maintained at RT in artificial cerebrospinal fluid (ACSF) containing (in mM) 125 NaCl, 2.9 KCl, 1.25 $KH_2PO4$, 26 $NaHCO_3$, 1 $MgCl_2$, 2 $CaCl_2$, and 20 glucose (pH 7.4) (with 95% $O_2$ and 5% $CO_2$ bubbling through the solution) for at least 1 hour prior to recording. After equilibration, individual slices were transferred to the recording chamber continuously perfused with ACSF (1 to 2 mL/minute). Recording micropipettes were pulled from borosilicate glass capillaries (1.5 mm O.D. Harvard Apparatus) for a final resistance of 3 to 6 MΩ and filled with a solution containing 135 CsCl, 4 NaCl, 0.4 GTP, 2 Mg-ATP, 0.5 $CaCl_2$, 5 EGTA, and 10 HEPES. The signals were recorded with an Axoclamp 700B amplifier (Axon Instruments, Union City, CA). Current and voltage clamp recordings were obtained from PyNs and ChCs in layers II/III of the ACC region. The cells in these brain regions were identified using a Nikon Eclipse FN-1 microscope with a 40× water-immersion objective and a DAGE MTI IR-1000 video camera. Neurons were visualized using IR-DIC to evaluate their orientation and morphology. ChC interneurons were identified by the red fluorescence from Nkx2.1-CreERT2/Ai14. Whole-cell patch-clamp recordings were performed with a high cell resistance (greater than 8 GΩ before break-in). sIPSC and mIPSC recordings in voltage-clamp configuration were acquired at 2 kHz fixing the voltage at −70 mV. The IPSCs were recorded in the presence of the NMDA receptor antagonist DL-2-amino-5-phosphonopentanoic acid (AP-5) at 50 mM and the AMPA/kainite antagonist 6-cyano-7-nitroquinoxaline-2,3-dione (CNQX) at 10 μM. For measurement of mIPSCs, 1 μM tetrodotoxin (TTX) was added to the perfusion solution to block synaptic responses dependent on the AP.

For studying the excitability and EPSCs, the CsCl in the internal solution was replaced by K-gluconate, and the signals were recorded with a low-pass filter at 10 kHz. Whole-cell patch-clamp recordings in current clamp configuration with a high cell resistance were obtained for PyNs and ChCs [81]. The neurons were characterized electrophysiologically by applying negative and positive current pulses of 10 pA and 1000 ms to calculate the firing frequency and positive pulses of 50 ms to measure the features for the single AP. EPSCs were measured in the presence of bicuculline (10 μM) while holding the resting membrane potential at −70 mV using K-gluconate internal solution. Access resistance was monitored throughout the experiment, and results were discarded if changes greater than 20% occurred. Peak events were identified automatically using Minianalysis (Synaptosoft) and visually monitored to exclude erroneous noise. The frequency, amplitude, and distribution of events were analyzed. One euploid neuron (1 out of 20) with mIPSC frequency 8.9 Hz was defined as an outlier by the Grubbs test and removed from quantification. Mean values were compared using the Student *t* test. To analyze evoked action potential frequency from neurons in acute brain slices, two-way ANOVA and Tukey's post hoc analysis were performed. Data are presented as mean ± SEM.

## Statistical analysis

Data are presented as mean ± SEM. Unless specified otherwise, comparisons of mean differences between groups were performed by one-way ANOVA for multigroup comparisons and post hoc Student $t$ tests for pair-wise comparisons. $P < 0.05$ was considered to be statistically significant. For all quantification, *: $p < 0.05$; **: $p < 0.01$; ***: $p < 0.001$; ****: $p < 0.0001$; ns: not significant ($p > 0.05$).

## Supporting information

**S1 Fig. Quantification of perisomatic GABAergic boutons on PyNs (related to Figs 1 and 4).** (**A**) Normalizing *DSCAM* gene dosage does not change the increased level of APP in Ts65Dn cortices. Shown are representative western blots (left) and quantifications (right) of neocortical samples from each indicated genotype. Each dot in the bar chart represents the sample from 1 mouse. (**B**) Representative confocal image of a PyN in layer II/III in the ACC. The soma is labeled by anti-GRASP1 (blue). The presynaptic active zones are labeled by anti-Bassoon (green). The GABA vesicles in presynaptic terminals of GABAergic neurons are labeled by anti-VGAT (red). The perisomatic Bassoon+ puncta that overlap with VGAT + puncta were quantified as GABAergic boutons (yellow arrowheads). The green arrowhead indicates the VGAT-independent perisomatic Bassoon+ puncta. (**C**) To define the soma region for quantifying perisomatic GABAergic boutons, we drew a line that is both perpendicular to the apical dendrite and tangent to the edge of the PyN nucleus and then quantified GABAergic boutons in the GRASP1+ area below the line. (**D**) The soma size of PyNs in the ACC is not affected in Ts65Dn or Ts65Dn:*DSCAM*+/+/− mice. The quantifications show the mean circumference of the soma. Each data point is the mean in a mouse. Statistical tests are one-way ANOVA for multigroup comparisons and post hoc Student $t$ tests for pair-wise comparisons. **: $p < 0.01$; ns: not significant ($p > 0.05$). Unless specified, mean ± SEM is shown in the figures. The data underlying this Figure can be found in https://doi.org/10.5281/zenodo.7714234. ACC, anterior cingulate cortex; APP, amyloid precursor protein; DSCAM, Down syndrome cell adhesion molecule; PyN, pyramidal neuron.
(TIF)

**S2 Fig. Genetic normalization of DSCAM levels rescues the excessive GABAergic boutons formed on the PyN somas in the somatosensory cortex in Ts65Dn mice (related to Fig 1).** (**A**) Representative images of perisomatic GABAergic boutons innervating PyNs in layer II/III of the somatosensory cortex of euploid (wild-type), Ts65Dn, and Ts65Dn:*DSCAM*+/+/−. The right panel in each genotype group is the magnified view of the regions boxed by dotted lines in the left panel. The soma and proximal dendrites of PyNs were labeled by GRASP1. Yellow arrowheads point to GABAergic boutons as indicated by Bassoon+ puncta that overlap with VGAT+ puncta. (**B**) Quantification of the number of perisomatic GABAergic boutons per PyN in the somatosensory cortex. Each data point in the chart represent the mean in 1 mouse. One-way ANOVA for multigroup comparisons and post hoc Student $t$ tests for pair-wise comparisons. ***: $p < 0.001$. The data underlying this Figure can be found in https://doi.org/10.5281/zenodo.7714234. DSCAM, Down syndrome cell adhesion molecule; PyN, pyramidal neuron.
(TIF)

**S3 Fig. Normalizing DSCAM levels rescues the increased sIPSCs in the PyNs in the Ts65Dn neocortex (related to Fig 2).** (**A**) Representative traces of sIPSCs from PyNs in layer II/III of the ACC in the euploid, Ts65Dn and Ts65Dn:*DSCAM*+/+/− brain slices. (**B, C**) Quantification of sIPSC frequency (**B**) and amplitude (**C**). Approximately 2–4 PyNs were recorded

for each mouse. A total of 6 euploid, 7 Ts65Dn, and 6 Ts65Dn:*DSCAM*+/+/− mice were analyzed. N: 19 for euploid, 19 for Ts65Dn, and 16 Ts65Dn:*DSCAM*+/+/−. (**D**) Representative traces of sIPSCs from PyNs in layer II/III of the ACC in euploid:*Lhx6*-Cre, Ts65Dn:*Lhx6*-Cre, and Ts65Dn:*Lhx6*-Cre:*DSCAM*$^{flox}$ brain slices. *DSCAM* gene dosage was normalized in GABAergic neurons in the Ts65Dn:*Lhx6*-Cre:*DSCAM*$^{flox}$ mice. (**E, F**) Quantification of sIPSC frequency (**E**) and amplitude (**F**). For each mouse, 2–4 PyNs were recorded. A total of 4 euploid control, 4 Ts65Dn, and 4 Ts65Dn:*Lhx6*-Cre:*DSCAM*$^{flox}$ mice were analyzed. N: 12 for euploid, 14 for Ts65Dn, and 17 Ts65Dn:*DSCAM*+/+/−. One-way ANOVA for multigroup comparisons and post hoc Student *t* tests for pair-wise comparisons. *: $p < 0.05$; **: $p < 0.01$; ***: $p < 0.001$; ****: $p < 0.0001$; ns: not significant ($p > 0.05$). The data underlying this Figure can be found in https://doi.org/10.5281/zenodo.7714234. ACC, anterior cingulate cortex; DSCAM, Down syndrome cell adhesion molecule; PyN, pyramidal neuron; sIPSC, spontaneous inhibitory postsynaptic current.
(TIF)

**S4 Fig. Excitability, firing rate, and sEPSCs of PyNs (related to Fig 2).** Quantification of electrophysiology parameters of PyNs in the ACC in euploid (gray), Ts65Dn (light blue), and Ts65Dn:*DSCAM*+/+/− (pink) brain slices. Data shown as mean ± SEM. Kruskal–Wallis test with post hoc Mann–Whitney tests for two-group comparisons, except for (G). *: $p < 0.05$; **: $p < 0.01$; ns: not significant ($p > 0.05$). (**A-F**) Quantifications of membrane potential (mV) (**A**), threshold (mV) (**B**), SAP amplitude (mV) (**C**), SAP half-width (ms) (**D**), the depolarization velocity of SAP (dv/dt) (**E**), and repolarization velocity of SAP (dv/dt) (**F**). Cell numbers: 19 for euploid, 18 for Ts65Dn, and 17 Ts65Dn:*DSCAM*+/+/−. (**G**) Curves showing the relationship between the average firing frequencies of evoked AP (Hz) and the currents (pA) in PyNs. Two-way ANOVA, Tukey's multiple comparisons test. ns: $p > 0.05$. (**H**) Rheobase (pA) of PyNs in (**G**). One-way ANOVA with post hoc Kruskal–Wallis test. ns: $p > 0.05$. N: 16 for euploid, 18 for Ts65Dn, and 17 Ts65Dn:*DSCAM*+/+/−. (**I, J**) Quantification of sEPSC frequency (**I**) and amplitude (**J**). For each mouse, 2–4 PyNs were recorded. The data underlying this Figure can be found in https://doi.org/10.5281/zenodo.7714234. ACC, anterior cingulate cortex; PyN, pyramidal neuron; SAP, single action potential; sEPSC, spontaneous excitatory postsynaptic current.
(TIF)

**S5 Fig. DSCAM does not regulate the number of GABAergic neurons in the neocortex (related to Figs 1, 3, 4, and 5).** (**A, B**) DSCAM overexpression in Ts65Dn mice does not affect the number of GABAergic neurons in the neocortex. Brain sections from P28 mice were immunostained with anti-PV. Representative images are shown in (**A**), and quantifications of the density of PV+ neurons are shown in (**B**). Each dot represents the value from 1 imaging field that is 252.4 μm (width) × 200 μm (length) × 5 μm (thickness). Images were collected from layer II/III of the ACC. Three fields in each mouse were randomly selected for imaging. A total of 5 euploid, 6 Ts65Dn, and 5 Ts65Dn:*DSCAM*+/+/− mice were analyzed. One-way ANOVA for multigroup comparisons and post hoc Student *t* tests for pair-wise comparisons. *: $p < 0.05$; ns: not significant ($p > 0.05$). (**C, D**) Loss of *DSCAM* does not affect the number of GABAergic neurons in the ACC. Representative images and quantifications are shown in (**C**) and (**D**), respectively. A total of 4 *DSCAM*+/− and 4 *DSCAM*−/− mice were analyzed. Student *t* test. ns: not significant ($p > 0.05$). The data underlying this Figure can be found in https://doi.org/10.5281/zenodo.7714234. ACC, anterior cingulate cortex; DSCAM, Down syndrome cell adhesion molecule; PV, parvalbumin; P28, postnatal day 28.
(TIF)

**S6 Fig. Quantification of ChC axon terminals and boutons (related to Figs 3 and 5). (A)** ChCs were sparsely labeled by tdTomato (red), and AISs of PyNs were labeled by immunostaining with anti-phospho-IκB (green). Confocal image stacks (0.3 μm z-steps for 100 steps) were maximally projected along the z-axis. A region of 120 μm (length) × 80 μm (width) with the cell body in the top middle was quantified. Cartridges and boutons that colocalized with AIS were quantified. Cartridge number was defined as the number of cartridges within this region. Cartridge length was defined as the distance from the first to the last bouton that colocalizes with the AIS in that cartridge. Bouton size is defined as the length of bouton in parallel to AIS. Interbouton distance is defined as the distance between 2 neighboring boutons. **(B, C)** Phospho-IκB and AnkG shows equal fidelity in labeling neocortical AIS. **(B)** AIS in layer II/III ACC was colabeled by phospho-IκB (green) and AnkG (red). Shown are maximal projection of confocal image stacks (1 μm z-steps X 7 steps). **(C)** Quantification of the percentage of AIS that is labeled by phospho-IκB and/or AnkG. A total of 308 AIS were quantified, among which 303 were colabeled by phospho-IκB and AnkG. The data underlying this Figure can be found in https://doi.org/10.5281/zenodo.7714234. ACC, anterior cingulate cortex; AIS, axon initial segment; ChC, chandelier cell; PyN, pyramidal neuron.
(TIF)

**S7 Fig. Quantification of the interbouton distance between neighboring boutons and the AIS length of PyNs in the layer II/III of the ACC (related to Fig 3). (A, B)** Neither the average interbouton distance between neighboring boutons nor the AIS length of PyNs was affected in the layer II/III of the ACC of euploid, Ts65Dn, and Ts65Dn:*DSCAM+/+/−*. One-way ANOVA for multigroup comparisons and post hoc Student $t$ tests for pair-wise comparisons. ns: not significant ($p > 0.05$). Each data point in the chart represent the mean of 8–10 PyNs in 1 mouse. **(C, D)** The average interbouton distance between neighboring boutons and the AIS length of PyNs were not significantly different between $DSCAM^{2j/+}$ (+/−) and $DSCAM^{2j/2j}$ (−/−) mice. Student $t$ tests. Each data point in the chart represent the mean of 4–5 PyNs in 1 mouse. The data underlying this Figure can be found in https://doi.org/10.5281/zenodo.7714234. ACC, anterior cingulate cortex; AIS, axon initial segment; DSCAM, Down syndrome cell adhesion molecule; PyN, pyramidal neuron.
(TIF)

**S8 Fig. DSCAM expression levels in the mutant mice used in this study (related to Figs 4 and 5).** Representative western blots and quantifications of protein samples collected from the somatosensory cortex of $DSCAM^{+/+}$ (+/+), $DSCAM^{2j/+}$ (+/−), and $DSCAM^{2j/2j}$ (−/−) mice. DSCAM protein was not detected in $DSCAM^{−/−}$ by western blotting **(A)**. The level of DSCAM protein in $DSCAM^{+/−}$ was about 81% of that in $DSCAM^{+/+}$ **(B)**. Student $t$ test. *: $p < 0.05$; ****: $p < 0.0001$. The data underlying this Figure can be found in https://doi.org/10.5281/zenodo.7714234. DSCAM, Down syndrome cell adhesion molecule.
(TIF)

**S9 Fig. Coupling between presynaptic terminal growth and synaptogenesis in ChC development (related to Figs 3 and 5). (A)** ChC cartridge length and bouton number are strongly correlated in both euploid and Ts65Dn genetic backgrounds. Each dot presents 1 ChC. For each mouse, 4–6 ChCs were analyzed. A total of 4 euploid, 5 Ts65Dn, 4 Ts65Dn:*DSCAM+/+/−* mice were analyzed. $n = 21$ for euploid (gray dots), 26 for Ts65Dn (cyan dots), 21 Ts65Dn: *DSCAM+/+/−* (red dots). $R^2$ and p are calculated for linear regression. The gray line indicates the trend line for gray dots, while the black line is that for blue and coral dots. **(B)** ChC cartridge length and bouton number are strongly correlated in *DSCAM+/−* and *DSCAM−/−* mice. Each dot presents 1 ChC. For each mouse, 4–6 ChCs were analyzed. A total of 4 *DSCAM +/−* and 4 *DSCAM−/−* mice were analyzed. N: 19 for *DSCAM+/−* (green dots) and 19 for

*DSCAM−/−* (yellow dots). (C) ChC cartridge length and bouton size show weak, yet significant, correlation in euploid and Ts65Dn background. Each dot presents 1 ChC. (D) The correlation between cartridge length and bouton number is impaired in *DSCAM*$^{-/-}$ mice. Each dot presents 1 ChC. $R^2$ is small in both *DSCAM*$^{+/-}$ and *DSCAM*$^{-/-}$ mice, suggesting that linear regression only explains a small fraction of the samples. The correction is insignificant in *DSCAM*$^{-/-}$ mice ($p > 0.05$). (E, F) ChC cartridge length and interbouton distance shows no significant correlation between any 2 genotypes tested. Each dot presents 1 ChC. The data underlying this Figure can be found in https://doi.org/10.5281/zenodo.7714234. ChC, chandelier cell; DSCAM, Down syndrome cell adhesion molecule.
(TIF)

**S10 Fig. DSCAM regulates ChC cartridge length and synaptogenesis in a dosage-dependent manner (related to Figs 3 and 5).** (**A**) Western blots showing DSCAM levels in the neocortex. The mouse neocortex was taken immediately after perfusion. (**B-E**) The correlation analyses between DSCAM expression level and ChC cartridge length (**B**), bouton number (**C**), bouton size (**D**), or interbouton distance (**E**). In (**B**), the DSCAM level of a mouse is plotted against the mean of the total cartridge length in the volume specified in **S1 Fig** in this mouse. The average cartridge length is calculated as the average value of 4–6 ChCs sampled in each mouse. A total of 4 euploid (gray dots), 5 Ts65Dn (cyan dots), and 4 Ts65Dn:*DSCAM*+/+/− (red dots) mice were analyzed. $R^2$ and p are calculated for linear regression. The data underlying this Figure can be found in https://doi.org/10.5281/zenodo.7714234. ChC, chandelier cell; DSCAM, Down syndrome cell adhesion molecule.
(TIF)

**S11 Fig. Excitability, firing rate, and sEPSCs of ChCs (related to Fig 2).** Quantification of electrophysiology parameters of ChCs in the ACC in euploid (gray), Ts65Dn (light blue), and Ts65Dn:*DSCAM*+/+/− (pink) brain slices. Kruskal–Wallis test with post hoc Mann–Whitney tests for two-group comparisons, except for (G). *: $p < 0.05$; ns: not significant ($p > 0.05$). (**A-F**) Quantifications of membrane potential (mV) (**A**), threshold (mV) (**B**), SAP amplitude (mV) (**C**), SAP half-width (ms) (**D**), the depolarization velocity of SAP (dv/dt) (**E**), and repolarization velocity of SAP (dv/dt) (**F**). Cell numbers: 13 for euploid, 9 for Ts65Dn, and 8 Ts65Dn:*DSCAM*+/+/−. (**G**) Curves showing the relationship between the average firing frequencies of evoked AP (Hz) and the currents (pA) in ChCs. Two-way ANOVA, Tukey's multiple comparisons test. Euploid vs. Ts65Dn: $p < 0.0001$ (****); Ts65Dn vs. Ts65Dn:*DSCAM*+/+/−: $p > 0.05$. (**H**) Rheobase (pA) from ChCs in (**G**). Cell numbers: 13 for euploid, 9 for Ts65Dn, and 9 Ts65Dn:*DSCAM*+/+/−. (**I, J**) Quantification of sEPSC frequency (**I**) and amplitude (**J**). For each mouse, 2–4 PyNs were recorded. Cell numbers: 12 for euploid, 7 for Ts65Dn, and 7 Ts65Dn:*DSCAM*+/+/−. The data underlying this Figure can be found in https://doi.org/10.5281/zenodo.7714234. ACC, anterior cingulate cortex; ChC, chandelier cell; DSCAM, Down syndrome cell adhesion molecule; SAP, single action potential; sEPSC, spontaneous excitatory postsynaptic current.
(TIF)

**S1 Raw Images. The original, uncropped, and minimally adjusted images supporting all blot and gel results reported in the figures and supporting information files (related to Figs 1, S8, and S10).**
(PDF)

# Acknowledgments

We thank Drs. Seth Blackshaw and Thomas Kim for the *Lhx6*-Cre mice. We also thank Drs. Roman Giger, Jun Wu, Andrew Nelson, Dawen Cai, Jonathan Flak, and Martin Myers for

sharing reagents or technical support, Miao He, Yongjie Hou, Pedro Lowenstein, Ken Inoki, Yukiko Yamashita, and Dawen Cai for helpful discussions.

## Author Contributions

**Conceptualization:** Hao Liu, Bing Ye.

**Data curation:** Hao Liu, René N. Caballero-Florán, Ty Hergenreder, Bing Ye.

**Formal analysis:** Hao Liu, René N. Caballero-Florán, Ty Hergenreder, Geng Pan, Paul M. Jenkins, Bing Ye.

**Funding acquisition:** Bing Ye.

**Investigation:** Hao Liu, René N. Caballero-Florán, Ty Hergenreder, Tao Yang, Paul M. Jenkins, Bing Ye.

**Methodology:** Hao Liu, René N. Caballero-Florán, Ty Hergenreder, Tao Yang, Jacob M. Hull, Ruonan Li, Macy W. Veling, Lori L. Isom, Paul M. Jenkins, Bing Ye.

**Project administration:** Bing Ye.

**Resources:** Kenneth Y. Kwan, Z. Josh Huang, Peter G. Fuerst, Paul M. Jenkins, Bing Ye.

**Supervision:** Paul M. Jenkins, Bing Ye.

**Validation:** Bing Ye.

**Visualization:** Bing Ye.

**Writing – original draft:** Hao Liu, René N. Caballero-Florán, Bing Ye.

**Writing – review & editing:** Ty Hergenreder, Ruonan Li, Lori L. Isom, Kenneth Y. Kwan, Z. Josh Huang, Paul M. Jenkins, Bing Ye.

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
