## [Editor Report · Decision Letter 0]

20 Jan 2023

Dear Bing, 

Thank you for submitting your revised manuscript entitled "DSCAM gene triplication causes excessive GABAergic synapses in the neocortex in Down syndrome mouse models" for consideration as a Research Article by PLOS Biology.

Your revisions have now been evaluated by the PLOS Biology editorial staff, and I'm writing to let you know that we would like to send your submission out for re-review.

Once your full submission is complete, your paper will undergo a series of checks in preparation for re-review. After your manuscript has passed the checks it will be sent back to the reviewers. To provide the metadata for your submission, please Login to Editorial Manager (https://www.editorialmanager.com/pbiology) within two working days, i.e. by Jan 24 2023 11:59PM.

Best wishes,

Roli

Roland Roberts, PhD

Senior Editor

PLOS Biology

rroberts@plos.org

---

## [Decision Letter · Decision Letter 1]

22 Feb 2023

Dear Bing,

Thank you for your patience while we considered your revised manuscript "DSCAM gene triplication causes excessive GABAergic synapses in the neocortex in Down syndrome mouse models" for publication as a Research Article at PLOS Biology. This revised version of your manuscript has been evaluated by the PLOS Biology editors, the Academic Editor, and three of the original reviewers.

Based on the reviews, we are likely to accept this manuscript for publication, provided you satisfactorily address the remaining points raised by the reviewers and the following data and other policy-related requests.

IMPORTANT: Please attend to the following:

a) Please address the remaining minor requests from reviewer #2 and #3.

b) Please address my Data Policy requests below; specifically, we need you to supply the numerical values underlying Figs 1AEF, 2BCEF, 3CEF, 4BD, 5BDE, 6BCEF, S1AD, S2B, S3BCEF, S4ABCDEFGHIJ, S5BD, S7ABCD, S8AB, S9ABCDEF, S10BCDE, S11ABCDEFGHIJ, either as a supplementary data file or as a permanent DOI’d deposition.

c) Please cite the location of the data clearly in all relevant main and supplementary Figure legends, e.g. “The data underlying this Figure can be found in S1 Data” or “The data underlying this Figure can be found in https://doi.org/XXXX”
https://zenodo.org/record/7554842

We expect to receive your revised manuscript within two weeks. 

*Published Peer Review History*

*Press*

Best wishes,

Roli

Roland Roberts, PhD

Senior Editor,

rroberts@plos.org,

PLOS Biology

DATA POLICY:

Regardless of the method selected, please ensure that you provide the individual numerical values that underlie the summary data displayed in the following figure panels as they are essential for readers to assess your analysis and to reproduce it: Figs 1AEF, 2BCEF, 3CEF, 4BD, 5BDE, 6BCEF, S1AD, S2B, S3BCEF, S4ABCDEFGHIJ, S5BD, S7ABCD, S8AB, S9ABCDEF, S10BCDE, S11ABCDEFGHIJ. NOTE: the numerical data provided should include all replicates AND the way in which the plotted mean and errors were derived (it should not present only the mean/average values).

We require the original, uncropped and minimally adjusted images supporting all blot and gel results reported in an article's figures or Supporting Information files. We will require these files before a manuscript can be accepted so please prepare and upload them now. Please carefully read our guidelines for how to prepare and upload this data: https://journals.plos.org/plosbiology/s/figures#loc-blot-and-gel-reporting-requirements

DATA NOT SHOWN?

REVIEWERS' COMMENTS:

Reviewer #1:

The authors provided extensive revisions to address some of the reviewer's concerns. Although I am fully convinced of the Editor's decision to pick up some experiments as unnecessary for the current paper, I would endorse the publication of the revised manuscript in PLoS Biology. 

Reviewer #2:

The authors have done a nice job responding to my previous concerns. I have only one strong suggestion. I appreciate the authors' perspective that the proximal synapses are likely to be dominated by basket and chandelier cells (that are also likely to be PV-positive). However, there are some proximal inputs from other cell types (even SST-INs make some proximal synapses). Moreover, the Lhx6-Cre line will label PV- and SST-expressing cells. I suggest the authors temper their statements in several locations to note that "perisomatic" synapses seem to be altered and may be causal to the observed phenotypes. However, alteration of other inputs can't be ruled out. I don't think this change dampens the most important conclusions and places the overall study conclusions on more rigorous footing.

Reviewer #3:

The authors have carried out a massive effort to address all of my concerns and most of the other reviewers' concerns. I do not have any further comments. The manuscript is solid and addresses a relevant question in the field which is not yet understood: the molecular mechanisms underlying the increase of inhibition in a Down Syndrome model. 

Minor

-Line 114, "…male loss-of-function" isn't it heterozygous instead of loss of function?

-Since postsynaptic markers are not used (e.g. Gephyrin), it will be more appropriate to name them boutons or inputs, in the text and figures.

---

## [Editor Report · Decision Letter 2]

14 Mar 2023

Dear Bing,

Thank you for the submission of your revised Research Article "DSCAM gene triplication causes excessive GABAergic synapses in the neocortex in Down syndrome mouse models" for publication in PLOS Biology. On behalf of my colleagues and the Academic Editor, Eunjoon Kim, I'm pleased to say that we can in principle accept your manuscript for publication, provided you address any remaining formatting and reporting issues. These will be detailed in an email you should receive within 2-3 business days from our colleagues in the journal operations team; no action is required from you until then. Please note that we will not be able to formally accept your manuscript and schedule it for publication until you have completed any requested changes.

Best wishes, 

Roli

Senior Editor

PLOS Biology

rroberts@plos.org